# Complexity and Entropy in Physiological Signals (CEPS): Resonance Breathing Rate Assessed Using Measures of Fractal Dimension, Heart Rate Asymmetry and Permutation Entropy

**DOI:** 10.3390/e25020301

**Published:** 2023-02-06

**Authors:** David Mayor, Tony Steffert, George Datseris, Andrea Firth, Deepak Panday, Harikala Kandel, Duncan Banks

**Affiliations:** 1School of Health and Social Work, University of Hertfordshire, Hatfield AL10 9AB, UK; 2MindSpire, Napier House, 14–16 Mount Ephraim Rd., Tunbridge Wells TN1 1EE, UK; 3School of Life, Health and Chemical Sciences, STEM, Walton Hall, The Open University, Milton Keynes MK7 6AA, UK; 4Department of Mathematics and Statistics, University of Exeter, North Park Road, Exeter EX4 4QF, UK; 5University Campus Football Business, Wembley HA9 0WS, UK; 6School of Engineering and Computer Science, University of Hertfordshire, Hatfield AL10 9AB, UK; 7Department of Computer Science and Information Systems, Birkbeck, University of London, Malet Street, London WC1E 7HX, UK; 8Department of Physiology, Busitema University, Mbale P.O. Box 1966, Uganda

**Keywords:** fractal dimension, heart rate asymmetry, permutation entropy, parameter tuning, paced breathing, resonant breathing, heart rate variability (HRV), complexity, software

## Abstract

Background: As technology becomes more sophisticated, more accessible methods of interpretating Big Data become essential. We have continued to develop *Complexity and Entropy in Physiological Signals* (CEPS) as an open access MATLAB^®^ GUI (graphical user interface) providing multiple methods for the modification and analysis of physiological data. Methods: To demonstrate the functionality of the software, data were collected from 44 healthy adults for a study investigating the effects on vagal tone of breathing paced at five different rates, as well as self-paced and un-paced. Five-minute 15-s recordings were used. Results were also compared with those from shorter segments of the data. Electrocardiogram (ECG), electrodermal activity (EDA) and Respiration (RSP) data were recorded. Particular attention was paid to COVID risk mitigation, and to parameter tuning for the CEPS measures. For comparison, data were processed using Kubios HRV, RR-APET and DynamicalSystems.jl software. We also compared findings for ECG RR interval (RRi) data resampled at 4 Hz (4R) or 10 Hz (10R), and non-resampled (noR). In total, we used around 190–220 measures from CEPS at various scales, depending on the analysis undertaken, with our investigation focused on three families of measures: 22 fractal dimension (FD) measures, 40 heart rate asymmetries or measures derived from Poincaré plots (HRA), and 8 measures based on permutation entropy (PE). Results: FDs for the RRi data differentiated strongly between breathing rates, whether data were resampled or not, increasing between 5 and 7 breaths per minute (BrPM). Largest effect sizes for RRi (4R and noR) differentiation between breathing rates were found for the PE-based measures. Measures that both differentiated well between breathing rates *and* were consistent across different RRi data lengths (1–5 min) included five PE-based (noR) and three FDs (4R). Of the top 12 measures with short-data values consistently within ± 5% of their values for the 5-min data, five were FDs, one was PE-based, and none were HRAs. Effect sizes were usually greater for CEPS measures than for those implemented in DynamicalSystems.jl. Conclusion: The updated CEPS software enables visualisation and analysis of multichannel physiological data using a variety of established and recently introduced complexity entropy measures. Although equal resampling is theoretically important for FD estimation, it appears that FD measures may also be usefully applied to non-resampled data.

## 1. Introduction

Nonlinear measures of complexity and entropy are used increasingly in the analysis of physiological signals [1,2,3,4]. For those researchers, particularly clinicians, who are not primarily computer scientists but wish to apply such measures in their own field, using a graphical user interface (GUI) package may be advantageous. CEPS (standing for ‘complexity and entropy in physiological signals’) is one such open-source GUI [4]. As first published, CEPS included ten methods of estimating data complexity and 28 entropy measures, using MATLAB as the programming language.

Another such open-source package, published more recently, not limited to MATLAB or one-dimensional data, is EntropyHub, with no complexity measures and 18 ‘base’ entropy methods (extending to more than 40 when cross-, multiscale, multiscale cross-, and bidimensional entropies are included) [5]. Earlier, such GUIs were reviewed elsewhere [4]. While preparing the current paper, we also encountered an ongoing review on fractal dimension estimators [6] and an open-source software library for nonlinear dynamics, DynamicalSystems.jl. [7] that includes estimators for fractal dimensions. We utilised some fractal dimension and time series complexity estimators from this library for further analysis and comparison.

In our first paper on CEPS [4], we demonstrated its use in the analysis of 5-min ECG RR interval (RRi), blood flow (from photoplethysmography, PPG) and respiration (‘breath-to-breath interval’, BBi or ‘PP’, for ‘peak-to-peak’) data collected during paced breathing from nine participants. We found that most of the complexity and entropy measures tested decreased significantly in response to breathing at 7 breaths per minute, when compared to baseline, normal breathing, differentiating more clearly than conventional linear, time- and frequency-domain measures between breathing states. In contrast, Higuchi fractal dimension (FD_H) increased during paced breathing. As anticipated, for all three data streams, complexity and entropy measures differentiated more clearly than conventional linear, time- and frequency-domain measures, between spontaneous and paced breathing at 7 BrPM (breaths per minute).

Here, as adumbrated in our earlier paper, we have extended this analysis in a second repeated-measures study, comparing measures for baseline, self-paced breathing and breathing paced at five different rates, not just one, and for a larger cohort (*N* = 44). We have also analysed how measures change with data length, and which are appropriate for data segments shorter than the five minute recordings usual in short-term heart rate variability (HRV) studies. Instead of using PPG data, we also explored tonic (slowly changing) electrodermal activity (EDA). In addition, after further literature review, there are now many more measures available in CEPS than when it was first published (currently more than 70 as against 10 complexity measures originally, and around 50 entropies). We have applied several of them here, focusing on other fractal dimensions in addition to FD_H (CEPS now includes some 22 FDs) and heart rate asymmetry (five ‘classical’ HRA measures and 11 derived from Poincaré plots). In addition, five measures from DynamicalSystems.jl are included in our analysis. As well as the introduction of many more measures, other changes to CEPS since its first version include the ability to modify data (normalisation, binarisation, interpolation, coarse-graining, addition of coloured and other noises, data segmentation with or without overlapping windows, resampling and detrending), as well as an analysis section for displaying plots and results from multidimensional or graphical measures (currently only two [8,9]. Other measures also in course of implementation include diffusion entropy [10,11], Emergence, Self-Organization and Complexity [12]. At the end of the article, we have included a list of the more than 200 abbreviations used.

### Objectives

Following on from those listed in our earlier paper, our objectives here are as follows:To conduct brief literature reviews on fractal dimension (FD) and HRA measures, and a more extensive review on resonance breathing.To use CEPS and DynamicalSystems.jl to analyse RRi, respiration and EDA data, and to compare results.To compare findings when using a variety of CEPS FD, HRA and measures based on permutation entropy (among others) to investigate whether there are marked differences between the effects of paced, self-paced and non-paced breathing on such physiological data—for example, which measures are most/least responsive to changes in breathing rate.To examine changes and agreement in key measures between baseline or self-paced breathing and optimal (or ‘resonance’) breathing, and explore questions such as ‘do people breathe naturally at their ideal rate?’To investigate the effects of parameter tuning on these measures in this context.To update the online CEPS ‘Primer’ and Manual to take changes in CEPS into account.To assess whether and which complexity and entropy measures applied to RRi and respiration data may be more effective at differentiating between resonance breathing and other breathing states than some of the more conventional HRV indices.To examine briefly whether age, sex, perceived stress (‘Distress’ and its converse, ‘Coping’), ‘Mindful awareness’ and two dimensions of interoceptive awareness (‘Noticing’, or awareness of body sensations, and ‘Attention regulation’, or the ability to sustain and control attention to body sensation), as well as a third dimension, ‘Self-Regulation’, may affect how CEPS measures reflect breathing state.To explore correlations within ‘families’ of measures, and between individual measures when applied to different data types (RRi, respiration and EDA).To investigate the effects of different data lengths on standard HRV and CEPS measures, with a view to determining the shortest data length that is feasible for use in further research on self-training methods of stress management.To explore how modifying the data in different ways (interpolation or deduplication, resampling, detrending, normalisation, multi-scaling, addition of noise) affects HRV and CEPS measures, and whether some of these methods may in fact compensate for the effects of shortening data length.In conclusion, to determine which measures are most useful for differentiating between resonance breathing and other breathing states, while also performing well for short data.

## 2. Materials and Methods

### 2.1. Literature Reviews

#### 2.1.1. Fractal Dimension (FD) and Heart Rate Asymmetry (HRA) Measures

##### Fractal Dimension

“ …estimating a fractal dimension is not an easy task. Focusing on only a single number can mislead. The best practice we feel is to calculate several versions of Δ, from different methods and with varying the parameters of each method (including the range of ε) and produce, e.g., a median of the results.” (Datseris et al. 2021 [6])

Fractal dimension (FD) is a ratio measure of irregularity or complexity, and for a curve can be thought of intuitively as an object too detailed to be one-dimensional, but too simple to be two-dimensional [13], so in principle will be between 1 and 2 in value. Background information on FD and some of the algorithms used in its estimation, as well as on multifractality, can be found in the updated 244-page CEPS *Primer on Complexity and Entropy*, downloadable as part of the CEPS 2 package on GitHub, the internet hosting service for software development [https://github.com/harikalakandel/CEPSv2/tree/master] (accessed on 20 January 2023).

Databases (PubMed, Google Scholar) were searched using “fractal dimension” AND [measure originator’s Name], without further examination. Results are shown in Table 1.

Over 5000 papers on ‘fractal dimension’ (FD) are indexed in PubMed, the first of these dating back to 1975, with 547,000 hits for FD in Google Scholar, including 1360 review papers since 2021. The most cited of these Wen and Cheong 2021 [25], 50 citations, is on using FD for analysis of complex networks rather than time series data but does describe several algorithms based on box-counting methods. The next most cited review (Henriques et al. 2020 [26], 42 citations) concerns FD for heart-rate time series data, but only mentions four algorithms: the usual ones by Katz and Higuchi, a variant of the box-counting method by Barabási and Stanley [27], and correlation dimension (D_2_). The latter, although it does provide a measure of FD, requires relatively long data samples for accurate estimation [28], so will not be considered further in this paper. A third review [29] includes several methods in addition to the box-count estimator, with code available in R [30]. A more recent and useful review of (mostly box-count) FDs, with code in Julia, is that by Datseris et al. (2021) [6], with the associated code available on GitHub [https://datseris.github.io/] (accessed on 20 January 2023).

Given the paucity of FD reviews including more than a handful of measures, but the large number of papers available on the topic, Google Scholar was used to search informally for studies using MATLAB code, and then the authors contacted with an invitation to provide code for CEPS. Of the 10 researchers contacted, three did not reply (their methods are not listed in the above Table 1); one preeminent Canadian researcher (Witold Kinser) hoped to be able to provide code for his Spectral and variance FD algorithms but was unfortunately unable to do so in the time available. The remainder very kindly provided code, and advice on its implementation.

Further information on the measures implemented—including basic algorithms—can be found in their published papers and in the updated CEPS *Primer on Complexity and Entropy* on the GitHub site under the directory ‘doc’ [https://github.com/harikalakandel/CEPSv2/tree/master] (accessed on 20 January 2023).

##### Heart Rate Asymmetry (HRA)

“The accelerations and decelerations of heart rate are well-defined physiological processes, even though the specific mechanisms that govern them are very complex. The widespread belief that it is the parasympathetic branch of the autonomic system which is responsible for decelerations and the sympathetic branch which is responsible for accelerations is only a first approximation and in reality, these processes are much more complex” (Mieszkowski et al. 2016 [31]).

Whereas time series data from linear systems generally exhibit ‘time reversibility’, in that their statistical properties are invariant regardless of the direction of time, time irreversibility (where statistical properties vary with temporal direction) is a common signature of nonlinear processes. It may occur, for example, in the EEG during epileptic seizures, whereas between seizures the EEG dynamic is more of a reversible linear process [32]. HRA is another example of time irreversibility.

HRA is a flourishing area of research and has been found even in neonates [33]. It is usually considered or defined in terms of unevenness in the distribution of points above and below the line of identity in the Poincaré plot, which indicates instantaneous changes in the beat-to-beat heart rate [34].

Heart rate may accelerate or decelerate, and HRV has been found to differ in phases of acceleration and deceleration. There are several measures of HRA, such as Porta’s, Guzik’s, the Slope and Area indices, but until recently none could estimate such asymmetries in heart rate variability. A further measure, the Asymmetric Spread Index (ASI), based on the Poincaré plot, was created to remedy this shortcoming.

Three frequently used measures of HRA were developed first: Ehlers’ index (EI) [35], Guzik’s index (GI) [36,37] and Porta’s index (PI) [38]. Karmakar et al. went on to develop a Slope index (SI) [39] and an Area index (AI) [40]. They also proposed redefinitions of EI, GI and PI to represent increasing patterns of increase or decrease in HR, not just instantaneous changes [41]. More recently, Rohila and Sharma [42] published the ASI. Another approach has been to use coarse graining (i.e., multi-scaling) of the data prior to using asymmetry indices [43,44,45]. A multiscale asymmetric Detrended Fluctuation Analysis (DFA) approach is also possible [31,46,47]. Further HRA measures have been based on the Standard Deviations of the Poincaré Plot scattergram along its minor and major axes, i.e., its short-term and long-term variance (SD1 and SD2, respectively), on the relative contribution of accelerations and decelerations to such variance (C1a, C1d; C2a, C2d), and on the Standard deviation of the interbeat intervals of normal sinus beats in the ECG (SDNN) (see Table 2).

Databases (PubMed, Google Scholar) were searched using “heart rate asymmetry” AND [Name, e.g., Porta, or “Slope Index”] without further examination. Results are shown in Table 2.

#### 2.1.2. Resonance Breathing and Vagally-Mediated Heart Rate Variability (vmHRV)

The distinct and unique mind-body relationship that exists via bi-directional communication between the heart and the brain has long been recognised, dating back to Darwin’s own experimentation in 1872 [50]. The importance of this relationship is coherently demonstrated by a decrease in heart rate variability (HRV) that occurs in various comorbid psychological diseases [51] and its association with central nervous function via cardiac control.

The measurement of vagal influence on HRV, i.e., vagally mediated HRV (vmHRV), has led to the acceptance of vmHRV as a trait indicator of cortico-cardiac control, psychophysiological adaptability [52,53] and autonomic regulation, with increased vmHRV leading to improvements in physiological health [54]. Vagally-mediated heart rate variability biofeedback (vmHRVBF) has thus been developed to encourage slow smooth sinusoidal breathing [55,56]. In turn, this results in large oscillations in heart rate. Such rhythmic synchronisation of heartrate to the respiratory system in ‘respiratory sinus arrhythmia’ (RSA) causes changes to the baroreflex (via baroreceptors), resulting in what is known as coherent or resonance breathing [57]. Resonance breathing has many psychological and physiological health benefits in both clinical and non-clinical conditions. Conditions for which resonance breathing may be helpful include but are not limited to asthma [58,59,60]; cardiac ill-health [61,62,63]; depression [64]; pain [64,65]; anxiety and stress [66]. As well as psychological and physiological health benefits, vmHRVBF improves performance in sport and in sporting and academic performance [65,67]. The benefits of vmHRVBF go beyond the initial treatment and can persist for up to three months post-treatment [68].

Studies on vmHRVBF postulate that resonance at 0.1 Hz oscillations in various physiological systems is achieved via breathing at a fixed rate of six breaths per minute [69,70]. It is argued that this frequency is instrumental in optimising both mental and physical health. This is due to physiological pathways involved in HRV providing a physiological feedback loop which is activated by breathing at this frequency, which may also enhance interoceptive awareness [69]. However, individualised resonance frequency rates are also used [71], and some experienced researchers recommend using several breathing rates to evaluate individual resonance frequencies [57]. To determine the ideal individual resonance frequency, the protocol developed by Lehrer et al. [58] recommends measuring adult breathing rates in decremental steps of 0.5 breaths per minute (BrPM), from 6.5 BrPM down to 4.5 BrPM.

Breathing rate is not the only important factor here. Longer exhalation compared to inhalation has been shown to result in higher RMSSD and HF-HRV than when either the inhalation/exhalation ratio is the same or when inhalation is greater than exhalation [72]. To aid relaxation, the inhalation/exhalation ratio may well be a vital factor in paced breathing trials [73].

### 2.2. Study Protocol

In order to demonstrate the utility of CEPS, this paper presents a subset of data taken from a larger on-going study and focuses on conventional and entropy measures of HRV derived from electrocardiogram (ECG) and respiration data. Figure 1 provides an outline schematic of the protocol.

#### 2.2.1. Resonant Breathing Rate Selection Using Paced Breathing

When conducting a resonant breathing assessment (RBA), a therapist will take into consideration a number of subjective and objective measures, including the participant’s comfort and compliance at a particular breading rate, as well as their ability to breath at the slower rates without any hyperventilation symptoms, such as light-headedness, dizziness or feeling that the heart is pounding. Lehrer et al. [58] have outlined the RBA protocol developed in their clinic and Shaffer and Meehan [57] have also codified the RBA protocol and recommend considering six HRV measures for the purpose of selecting the resonance breathing rate (RBR). We have endeavoured to replicate the Shaffer and Meehan [57] protocol in the current study, using the following measures: (1) Low Frequency (LF, (0.04–0.15 Hz) Absolute Band Power, in units of ms^2^ (LFBP); (2) power of the LF Spectral peak, in units of ms^2^ (peak low frequency power, or PLFP); (3) normalised LF HRV, in normalised units (nu); (4) peak-to-trough difference in heart rate (HR), or ‘HRMaxMin’ (in beats per minute); (5) Phase Relationship of HR to Respiration rate (in degrees); and (6) comfort level. Shaffer and Meehan [57] also discussed how to select the RBR and how to ‘Break Ties’ when different breathing rates score maximally on different measures. This does, however, require a degree of clinical judgement.

Thus, in order to circumvent any issues with a subjective selection of the RBR, the breathing rate with the highest LF Spectral peak was chosen for each person. This was considered acceptable because participants were closely monitored for comfort and compliance during the trials and people who failed to breathe at the paced rate or had a significant number of artifacts were excluded from the data analysis (three people). When comparing RBR selection based purely on the trial with the highest LF Spectral peak against the more involved ‘clinical’ selection method outlined in the papers above, results differed for only five people out of 44 (11%), with three participants having a trial 0.5 BrPM faster and two 0.5 BrPM slower for the ‘LF Spectral peak’ than for the clinical selection method.

Although in theory approximately 1 min of data are needed to assess the HF components of HRV, and approximately 2 min for the LF component, it is generally accepted that for accurate estimation of short-term HRV, five minutes of data are required, and that frequency-domain measures should be preferred to time-domain measures [74]. We followed these guidelines here. For some time-domain measures, however, such as SDNN and RMSSD, even 10 s of data may be useful [75].

#### 2.2.2. Ethics

Ethical approval was granted from the Open University’s Ethics Committee (Project number: HREC/4117/Steffert/Banks). Participants gave written consent, and each gave permission for their anonymised data to be stored on the Open University ‘Open Research Data Online’ (ORDO) Open Access repository database (https://ordo.open.ac.uk/) (accessed on 20 January 2023).

##### COVID Risk Mitigation

It was not possible for the data to be collected during this study without the use of specialist equipment and measurements that necessitated proximal interaction with participants. The study coincided with the COVID pandemic and as a result we were required to mitigate the risk of exposure to the virus for both participants and researchers. The protocols as laid out in this application conformed to both the UK Government and the updated Open University (OU) Guidelines on face-to-face contact between individuals. Students from the University Campus of Football Business (UCFB, Wembley) who were participants in the early part of this study were divided into suitably sized bubbles which remained separated from others during term time and during the whole time that the experiments were run. UCFB required their students to undergo a polymerase chain reaction (PCR) test at the start of term and thereafter twice weekly lateral flow tests. Each of the venues (UCFB, Wembley or OU Laboratories) had ample hand sanitising gel and handwashing facilities to limit surface contamination.

A maximum of two researchers at any one time operated in separate laboratories, underwent lateral flow tests twice weekly and only proceeded if the tests were negative. Each of them was required to have been doubly vaccinated, wear face masks throughout the procedures and, unless placing electronic sensors on the participants, kept a safe distance of two metres. In all cases the researchers and participants stated at the start of each session that neither they nor any member of their households (a) were considered clinically extremely vulnerable to COVID; (b) had not recently acquired any new COVID symptoms; (c) had not encountered anyone who had tested positive for COVID in the previous 10 days. Each venue had a named individual who was responsible for monitoring adherence to the health risk assessment and COVID transmission risk control measures. We appreciated that the virus had brought with it health and safety risks, and additional concerns such as personal stress and anxiety. Information about COVID and the support systems for personal stress and anxiety at UCFB, Wembley or OU was in place for all staff and students to alleviate these concerns.

The physiological recording equipment was of clinical grade and as such conformed to the necessary Health and Safety requirements for human experiments. Disposable electrodes were used, and the chest straps used to measure respiration were placed over the clothes and did not come into contact with the skin of participants. The research team had a contingency plan should COVID incidence increase very significantly over the course of the research necessitating reassessment of the project risks. This included the verification of COVID status of researchers using the COVID.status app developed by International Health Codes Ltd. (London, UK) (https://internationalhealthcodes.com/) (accessed on 20 January 2023) that recorded lateral flow COVID tests on a mobile phone. COVID status could be verified and shared using quick response (QR) codes.

#### 2.2.3. Participants

Participants (*N* = 44, Female = 18) were recruited by e-mail or word of mouth from the students and staff of University College Football Business (UCFB, *n* = 21) and via snowball sampling from other venues. Participants were sent a study participant information sheet. Participants selected their relevant age from seven age bands (Table 3).

On arrival for their single study session at UCFB and other approved venues, the exclusion/inclusion criteria were verified. Participants who had a pacemaker or other cardiac issues, breathing-related difficulties (e.g., COPD (Chronic Obstructive Pulmonary Disease), Emphysema or asthma), Raynaud’s disease, post-traumatic stress disorder (PTSD), or were on Beta blockers were excluded, as these can affect the heart rate variability. Participants were reminded of the purpose of the study and its ethical framework, including their right to withdraw from the study at any time. Strict adherence to Governmental and institutional COVID 19 protocol was maintained throughout.

#### 2.2.4. Data Collection

Participants completed three questionnaires including the ten-item Perceived Stress Scale (PSS) [76], with its Coping and Distress subscales, and the Mindful Attention Awareness Scale (MAAS) [77], abridged from fifteen items to five [78]. They also completed the Multidimensional Assessment of Interoceptive Awareness (MAIA) [79], consisting of fifteen items taken from three subscales: Noticing (four items), Attention Regulation (seven items), Self-Regulation (four items), omitting those subscales not relevant to the study. Given the purpose of the current paper, the questionnaire findings will be reported in a follow-up study.

Participants were instructed to sit comfortably and try not to move or talk during seven 5 min 15 s recordings. During the first baseline trial they were instructed to breathe normally with their eyes open and to avoid meditation, deliberate slow breathing, or falling asleep. Next, a self-paced slow breathing trial was conducted, in which participants were instructed to “breathe in a slow and comfortable manner, whatever is best for you; don’t try too hard but do try to maintain the slow and comfortable breathing for the full 5-min recording”.

Five slow paced breathing trials were then presented in ‘block random’ order (randomised using the ‘getRandList’ function from the ‘randomizeR’ package in RStudio). Participants were instructed to look at an onscreen display and follow ether the “pacer” line or the bar graph (see Figure 2), by breathing in as the line or bar went up and breathing out as they went down. Breathing was paced at 7, 6.5, 6, 5.5 or 5 breaths per minute.

A Nexus-10 physiological acquisition amplifier and BioTrace+ software (MindMedia, Herten, The Netherlands) was used to collect physiological data on a dual screen laptop. ECG data were sampled at 256 Hz, PPG data at 128 Hz, and EDA and Respiration data at 32 Hz, and saved for export at 256 Hz. The participants were seated comfortably in front of a second external monitor, which displayed the onscreen respiration pacer. Three self-adhesive electrodes were attached to their wrists to collect their ECG signal. A PPG pulse sensor and two electrodermal activity (EDA) electrodes were fitted to the fingers of the non-dominant hand (usually the left) and an abdominal respiration belt was fitted over the clothes at the level of the umbilicus (‘belly button’). A small consumer-grade ‘KYTO’ Bluetooth PPG sensor (KYTO Fitness Technology, Dongguan, China) was also positioned on an ear lobe (results not presented in this paper).

Participants were told to use slow diaphragmatic or “belly” breathing, avoid inflating their chest or holding their breath. The breathing pacers were set at a 40/60 inhalation to exhalation ratio, meaning the inhalation was shorter than the exhalation. They were told, “you should not try too hard, just take normal but slow breaths in through your nose and out through your mouth. If you get dizzy or uncomfortable, this is a sign of breathing too deeply or shallowly, so just breathe normally. We can stop at any time if you need to”.

Some participants found it difficult to concentrate on the pacer for 5 min and several started to fall asleep. In such instances the researchers gently prompted the participant to stay on task.

After each trial, participants were asked to rate on a scale from zero to 10: “How easy or difficult was it to breath at that rate?” (“Very Difficult” = 0 and “Very Easy” = 10). Further questions were: “How comfortable did you find that breathing rate?” (“Very Uncomfortable” = 0 and “Very Comfortable” = 10); “How accurately do you think you did that breathing rate?” (“Very Inaccurate” = 0 and “Very Accurate” = 10); and “Did you have any discomfort or dizziness?” (“None” = 0 and “Very dizzy/uncomfortable” = 10).

After their session, each participant received a personalised report of the data used to identify their resonance breathing rate and indicating which was their ideal resonance rate.

#### 2.2.5. Software and Data Processing

##### Updating CEPS for This Project

In our first study [4], we compared values of FD_H, multiscale permutation entropy (mPE) and conventional HRV measures LF and HF relative (%) power, for both ‘normal’ breathing and breathing paced at 7 BrPM. Here, we wished to broaden our analysis and consider three ‘families’ of measures in addition to those from conventional HRV: fractal dimensions more generally, further measures based on permutation entropy, and heart rate asymmetry metrics (see above, Section 2.1.1). As listed in Table 4, we have also implemented, or are still in the process of implementing, a number of other measures in CEPS. The Table also includes a selection of other measures we plan to include in the future if circumstances permit.

As before, codes for measures to be included in CEPS were requested from those who wrote them or obtained from the academic papers in which they were published. The code for Ehlers’ Index was written by Deepak Panday, and for the other HRA measures was generously provided by Ashish Rohila [https://www.linkedin.com/in/dr-ashish-rohila-87739255/] (accessed on 20 January 2023).

Further information on the HRA measures implemented can be found in the published papers referenced above and in the CEPS *Primer on Complexity and Entropy*, which is downloadable as part of the CEPS 2 package [https://github.com/harikalakandel/CEPSv2/tree/master] (accessed on 29 January 2023).

Following implementation in CEPS, a test dataset and results were then sent to the code’s originators so that the codes as implemented could be verified. Not all who were contacted responded to our requests, so some measures could not be implemented, and others could not be verified.

##### Comparison with Estimators from DynamicalSystems.jl

DynamicalSystems.jl [7] is a general-purpose software library for nonlinear dynamics and nonlinear timeseries analysis. It does not offer a GUI interface like CEPS. Rather, it is a formally a package for the Julia programming language, that can be loaded in a scripting environment. Once loaded, it provides several hundred algorithms for calculating quantities typically relevant in nonlinear dynamics, e.g., estimators for fractal dimensions or other complexity measures such as permutation entropy. Recently in has been used in an extensive review of fractal dimension estimators [6].

Here we used Δ and Δ_2_, fractal dimension estimators from DynamicalSystems.jl and two timeseries complexity estimators. The fractal dimension Δ is fundamentally different from the time series FDs considered so far in this paper. Instead of quantifying the ‘roughness’ of the graph of a function (like the Higuchi estimator), Δ quantifies the effective dimensionality of the underlying dynamics. Specifically, we first analyse each time series using the approach of [118] to estimate an optimal delay embedding that most accurately represents the underlying dynamical attractor representing the dynamics generating the data. Once that is estimated, we reconstruct the attractors by delay embedding the time series. On this higher-dimensional object we use the well-established Grassberger-Procaccia algorithm [119] to estimate a fractal dimension as the scaling of the correlation sum versus a size parameter. Notice that while typically the reconstructed attractor would be higher than 2-dimensional, here we purposefully only embed up to two dimensions, to force the fractal dimension into the interval (1, 2), as used for the Higuchi dimension (to enable a simpler numerical comparison across the two methods). Unbounded values of Δ were also computed, and two optimal delay times, tau and tau2. In DynamicalSystems.jl, the Higuchi dimension was computed using values of k from 2 to 256, exponentially spaced, and the resulting values averaged. The other two complexity measures we used from DynamicalSystems.jl are wavelet entropy (‘wavent’) [117] and permutation entropy (‘perment’, or PE, with order m = 3 or 4 and lag as either 1 or the least mutual similarity time of the timeseries) [120]. Both measures were chosen because they are suitable quantifiers of complexity of timeseries and useful in classification tasks (such as the ones we attempt here), but also because they have been shown to be effective even with very short or non-stationary time series lengths (which we also have here). FD_H was also computed.

Results from DynamicalSystems.jl and CEPS were compared, for both FD_H and for PE with m = 4 and lag = 1. Many other entropic time series methods are included in DynamicalSystems.jl, but not in CEPS. A partial list can be found in [7], with a fuller list in the software documentation.

##### Other Software Used

Data were exported from BioTrace+ and then pre-processed in RStudio. The raw ECG were analysed using Kubios HRV Premium 3.1.0. (Kubios Oy, Kuopio, Finland). As in our previous study, respiration intervals were extracted from raw data using ProcessSignals, a package still in development [121]. For both the RRi and respiration data, artefacts were removed using a mix of manual and automatic methods.

Because Kubios HRV does not offer a batch processing option, Meg McConnell kindly agreed to process the segmented and interpolated RR interval data, using her Python package RR-APET, which has been evaluated against Kubios HRV [122].

The Kubios and CEPS outputs were analysed statistically using IBM SPSS Statistics v26 and RStudio (2022.7.1.554) [123]. Several subsidiary R packages were also used, in particular the ‘durbinAllPairsTest’ from the PMCMRplus package (1.9.3) [124] and ggstatsplot (0.9.1) [125]. Further analysis was carried out in Microsoft^®^ Excel^®^ 2019.

#### 2.2.6. Data Processing

In our analysis, 31 files were excluded because of excessively noisy data (mostly attributable to slippage of the respiration sensor belt), or where breathing peaks and troughs were otherwise difficult to identify (two participants were already excluded because of recording difficulties and data loss). After excluding noisy data, and with further processing and artefact removal, 308 files remained. Using the Panday-Kandel ‘ProcessSignals’ package, as in our first study, inbreath [End(n)_to_peak(n + 1)] and outbreath [Peak(n)_to end(n)] durations were calculated, as well as their ratio [Out(n)_to_In(n + 1) ratio] and peak-to-peak durations [peak(n)_to_peak(n + 1)] (Figure 3).

#### 2.2.7. Data Pre-Processing and Modification

The following data pre-processing and modification methods now available in CEPS 2 were used:

##### Detrending

Over the course of each 5–minute trial, EDA tended to decrease. This would render measures of FD meaningless [126] (p. 82), so data were both deduplicated and detrended before further analysis using nonlinear measures from CEPS or dynamicalsystems.jl (ECG data were detrended in Kubios HRV, and RSP data were not detrended).

##### Data Segmentation (‘Cut Files’)

In addition, 5-minute recordings were segmented in two different ways:(a) divided into 1-min, 2-min, 3-min and 4-min slots, all starting at the same time point, in order to analyse the effects of data length, and (b) dividing the data into five equal segments in order to investigate changes over time during each trial. These two methods were used on both the RR interval (RRi) data, extracted from the Kubios HRV output files after artefacts were removed, and on the deduplicated and detrended EDA data.

##### Adding Noise (‘Add Noise’)

Adding white or coloured noise to a weak signal can sometimes, paradoxically, increase its detectability through the process of stochastic resonance [127]. It can also be used as an up-sampling method of data augmentation to enhance classification in machine learning [128]. Different coloured and white noises can now be added to signals in CEPS, or used as stand-alone data to test how noises affect the different measures.

##### Interpolation

Short data were interpolated using one of two methods—linear or ‘nearest-neighbour’—and a recursive ‘finegrid’ method [129], with 1, 2 or 3 finegrid points for 1-min data, 1 or 2 points for 2-min data, and 1 point for 3-min data (4-min data were not interpolated).

##### Equal Resampling, Using ‘Shape-Preserving Piecewise Cubic Spline Interpolation’

“We think we measure [Time] out with clocks, but in fact … it measures us out by events” (Sybil Marshall [130], p. 110)

When used on time series data, the algorithms in DynamicalSystems.jl will only produce correct results if the data are sampled with a constant sampling rate [14,126] (p. 93). By definition, ‘interval’ data such as RRi, are defined by physiology, not by the clock, and are not necessarily uniformly resampled, although they may need to be to give correct results for some CEPS measures. Here we used the ‘resample’ function from the MATLAB Signal Processing Toolbox [131] with 4 Hz and 10 Hz as resampling rates for the RRi data, comparing results for RRi data that were not resampled.

(Deduplication of up-sampled EDA and respiration data was carried out separately in MATLAB.)

#### 2.2.8. Parameter Selection

Parameter selection was conducted in CEPS to fine-tune results. This was found to be crucially important.

Parameter tuning is not necessarily a precise and rigorous process. The optimal parameter for one application (e.g., determining the maximum value of a measure) may not coincide with the best parameter for a different situation, as here, where parameters were selected to maximise differences in value between breathing trials. Taking distribution entropy (DistEn) as an example, a measure which is purportedly rather independent of embedding dimension (m) [132], at baseline the value of m that provided the maximum value of DistEn was 2 for 20 of the 44 study participants and 3 for 17 others, whereas during resonant breathing (RBR), the maximum value of DistEn occurred for 33 participants with m = 2. and for only seven with m = 3. In contrast, the maximum difference in DistEn between baseline and RBR occurred for 25 participants with m = 1, but only for one participant when m = 2, and for two with m = 3. The parameters for optimising values and maximising differences are clearly divergent.

In our first study [4], we selected the parameter *k*_max_ for FD_H based on a very small sample (*N* = 9) and found that the measure differentiated reasonably well between normal and paced breathing with *k*_max_ = 5, or for *k*_max_ between 9 and 14. Here, we went on to use several strategies to tune parameters:

(1) Initially, from a larger, convenience sample of 70 recordings from two study participants, we determined optimum parameters using a method based on the robust coefficient of variation (RoCV) of the data, Unfortunately, for FD_H, this resulted in a value of *k*_max_ that no longer showed a significant difference between the Baseline and RBR Trials, as would have been expected from our previous study.

Therefore, other strategies were then explored:

(2) Setting the parameters that provided the greatest RoCV in FD_H for the whole group (*N* = 44).

(3) Selecting the parameters that provided the greatest difference between the median values of FD_H at baseline and during resonance breathing for the whole group.

(4) Tuning parameters according to the median of the differences in FD_H at baseline and during resonance breathing across all participants.

(5) Determining the parameter according to the number of maximal differences between baseline and RBR for a range of parameters, again for the whole group.

A final decision on the parameter setting to use was then arrived at, on the basis of results using all the above four methods (2–5). Of course, if they agreed, that was a simple matter. However, they did not always do so (Appendix A) (a link to these online materials is provided below). If they did not, a further method was used; counting the number of positive and negative differences between Baseline and RBR or 5 BrPM and taking their ratio. The method for which this was farthest from 1.0 was also taken into account. When these five methods still did not provide an unequivocal answer, the plot of the measure against the parameters concerned was taken as the final arbiter.

This entire process was very time-consuming, and it also soon became clear, especially from method (5), that parameters could not be selected unequivocally for all participants, that any selection would be group-specific and not necessarily generalisable to other cohorts, and that parameters that could be selected to maximise one result (here, the difference between baseline and RBR) might not be relevant for other comparisons (e.g., between baseline and breathing at 5 BrPM). For many measures, therefore, determining the most appropriate parameter/s to use was a matter of compromise. Ideally, an automated method of parameter tuning could be developed for use in different contexts, in order to speed up the process. This is currently beyond the scope of CEPS but is a future possibility.

#### 2.2.9. Statistical Analysis

##### Data Distribution

Data distribution was explored using the Shapiro-Wilk test.

##### Analysis of Variance 1. Welch’s ANOVA

In an initial exploration of the data (*N* = 7), Welch’s ANOVA (analysis of variance) was used as a robust test of equality of means, together with post-hoc Bonferroni (Dunn) tests, to assess differences in CEPS measures with the various breathing rates and the effects of different interpolation methods (type of interpolation and number of interpolated data points) on these measures. A significance threshold of 0.01 was taken for the post-hoc adjusted *p*-values, with *p*-values of 0.01 or above being taken as not significant. Bootstrapping was not used.

##### Analysis of Variance 2. Friedman Tests, Kendall’s W and Conover Tests

As Welch’s ANOVA requires normality of distribution, from this point on Friedman tests were carried out for nonparametric one-way repeated measures analysis of variance by ranks, with Bonferroni post hoc tests, and using Kendall’s W (coefficient of concordance) as a measure of effect size (ES), for which 0.1 is considered small, 0.3 medium, and greater than 0.5 as strong [133,134]. To reduce numbers of false positive findings, a significance threshold of 0.001 was taken for the Friedman tests (i.e., *p* < 0.001), with post-hoc adjusted *p*-values of ≤0.001 considered significant. This approach was taken in preference to the more accepting Benjamini-Hochberg procedure used in our previous paper [4]. However, using Kendall’s W does not guarantee that all pairwise comparisons are significant; some may be highly significant and others much less so [135]. To further refine results, the Conover test, another post-hoc test for non-parametric data in a within-subjects study design [136], was also used, and the results ‘top-sliced’ so that only those measures for which the standardised value of the Conover statistic was ≥ 0.8 were considered further.

In addition, counts were made of the numbers of participants for whom each measure increased or decreased between the Baseline and RBR trials. If the ratio of increases to decreases was greater than 0.795 or less than 0.205 (i.e., with 35 or more increases, or 9 or fewer increases in our study group of 44 participants), we took this to indicate that the measure changed in the same direction for most of the group. (The Binomial test would allocate a *p*-value of < 10–3 to a ratio of 0.795.)

##### Assessing Agreement. Intraclass Correlation Coefficients (ICCs) and Simple Correlations

Intraclass correlation coefficients (ICCs) were calculated in SPSS (Statistical Package for Social Science) to assess agreement between the values of each measure for segmented data of different durations (1 to 5 min). Lack of agreement could indicate non-stationarity of data [6], or that short segments did not provide sufficient information to permit accurate calculation of the measure. A two-way mixed model was used, aiming for consistency rather than absolute agreement, and the ‘ANOVA with Friedman’s Test’ option was selected. In addition, Spearman’s *rho* was computed for correlations between the value of each measure for five minutes of data and its values at shorter durations, and also for correlations between measures in difference trials (baseline or self-paced vs. 5 BrPM or RBR) or computed for different data types (RRi, respiration intervals and EDA). Data were not analysed for outliers, homoscedasticity (homogeneity of variances) or normal distribution of residuals [137], so a formal (linear) regression analysis was not undertaken. As for the Conover statistic, results were ‘top-sliced’ so that only those measures for which the standardised value of the ICC was ≥ 0.8 were included in further analysis.

##### Combining the Results of Conover Tests and ICCs

Based on the two previous steps, those measures that were found to best differentiate between the trials, as well as those that best showed agreement across durations, were tabulated, and plotted.

This was done for the following datasets:CEPS and RR-APET measures for non-resampled RRi data;CEPS and DynamicalSystems.jl measures for RRi data resampled at 4 Hz;CEPS measures for RRi data resampled at 10 Hz;CEPS and DynamicalSystems.jl measures for detrended and deduplicated EDA data.

The raw respiration and breathing interval data were analysed only using Conover tests, not ICCs.

##### Effects of Age, Sex, Perceived Stress and Other Trait and State Measures

The effects of age, perceived stress, ‘Mindful awareness’ and the two dimensions of interoceptive awareness on how CEPS measures reflect breathing state were assessed. Methods and Results for this Section are presented in the Appendix A.

##### Correlations within ‘Families’ of Measures, and between Individual Measures

Spearman’s *rho* rather than Pearson’s *r* was also used to explore correlations within ‘families’ of measures, and between individual measures when applied to different data types (RRi, respiration and EDA). Methods and Results for this Section are again presented in the Appendix A.

## 3. Results

### 3.1. Normality of Data

Data for over 40% of the Kubios HRV measures were not normally distributed, with around 36% of the CEPS ECG RRi measures also not normally distributed. Almost 68% of the CEPS EDA measures were not normally distributed, while none of CEPS measures for the raw respiration data and only 18% of those for the breathing interval data were normally distributed.

### 3.2. Data Resampling and Modification

#### 3.2.1. The Effects of Data Resampling on CEPS Measures

For some (but not all) trial pair comparisons, resampling RRi data at 4 Hz marginally increased the numbers of the Conover statistic above the 95th percentile when compared with the non-resampled data, as shown in the Figure 4 below (a comparison with results for data resampled at 10 Hz is not shown, for reasons explained below).

For the 5-min RRi data, the 95th percentile of the Conover statistic was calculated for the following grouped types of measures: (1) 22 fractal dimensions (FDs); (2) 40 heart rate asymmetries and those derived from Poincaré plots (HRA); (3) 8 measures based on permutation entropy (PE); (4) 19 measures based on recurrence quantification analysis (RQA); (5) 51 other complexity measures (OC); and (6) 54 other entropies (OE). Numbers of the PE and RQA measures varied slightly between the noR and 4R groupings, and for the OC and OE measures between the 10R and other groupings (Table 5).

Note that the 95th percentile was greater for the non-resampled RRi data than for the data resampled at 4 Hz, for all except the RQA and OE types, and that while resampling at 10 Hz improved results for the RQA, OC and OIE measures, it did not do so for the others. Given that increasing the resampling rate to 10 Hz did not change the finding in Figure 4 at all, and that it was far more computationally demanding than resampling at 4 Hz, it was not investigated further in this study in any depth.

#### 3.2.2. The Effects of Data Modification—Mitigating for the Effects of Data Segmentation (Shortening)

Initial use of Welch’s ANOVA on a pilot sample of RRi data (*N* = 7) provided a provisional indication of which CEPS measures are most/least likely to be sensitive to data interpolation, and which most often showed significant differences with breathing rate for the interpolation methods used; mPE5, mLZC7 and PJSC with default parameters appeared most useful; CCM, RCmDE3 and Kurtosis the least useful.

For the full dataset (308 trial recordings, 44 participants), several measures, including DistEn, Edge PE (EPE) and FD_C, showed significant differences in RBR (‘best’) vs. the other breathing rates (‘rest’), with *p* < 0.05 (using the Mann-Whitney test). Of these, DistEn and FD_C showed effect sizes (*Z*/√*N*) for the interpolated 3-min data that were in fact marginally *better* than those for the full 5-min data (0.256 vs. 0.233, and 0.302 vs. 0.275, respectively). However, improvements over results with the non-interpolated data were in general minimal and inconsistent over the different data lengths, and effect sizes remained small. Out of 17 measures investigated, the strongest effect size was for FD_C.

Comparing the interpolation methods tested for the 17 CEPS measures and three data lengths (1-, 2- and 3-min), ‘L1′ (the linear method using a single point) most commonly gives the highest ES (23 occurrences), followed by L2 (7 occurrences), and finally N1 (‘nearest-neighbour’ method using 1 point) (4 occurrences), with no occurrences for N2. For the interpolated measures with *p* < 0.001, 19 were for linear interpolation (median ES 0.235), and only eight for the nearest-neighbour interpolation method (median ES 0.228). Interpolations of 2 points or more were not as useful for classifying ‘best’ vs. ‘rest’ as interpolations of 1 point; 14 ‘L’ (0.237 vs. 0.226 for 1 vs. 2 points interpolated); 8 ‘N’ (0.229 vs. 0.226 for 1 vs. 2 points interpolated). Thus, if interpolation is to be used to increase data length, it might be appropriate to use 1-point linear interpolation, but there is no guarantee that this will improve classification. Indeed, overall, and for several measures such as FD_H, mPE1, mPM_E, RPE and TPE, the effect of interpolation appeared to be to *reduce* effect size, not increase it.

In general, for the CEPS measures tested (a mix of FDs, Poincaré-derived measures, permutation, and other entropies), 1-point linear interpolation provided the best differentiation between breathing trials, and linear interpolation better than nearest-neighbour interpolation. However, for FD_P and FD_M the number of points interpolated appears to be less of an issue, and for FD_C and AttnEn, the interpolation method may not materially affect results either, indeed, for CoSiEn, results may be better using nearest-neighbour than linear interpolation (details available on request).

The effects of interpolation on standard HRV measures were also assessed when comparing the RBR with other breathing rates, using results generated by Meg McConnell’s Python-based software package, RR-APET [122] and Mann-Whitney tests, as for the CEPS measures. Here, marked improvements were found for the 3-min LF percentage power based on the Lomb-Scargle periodogram, using both ‘L1′ and ‘N1′ as interpolation methods (ES = 0.307 and 0.309, respectively); ES for the original 5-min data was only 0.234), For LF peak frequency (based on the Welch periodogram), improvement was from ES = 0.244 (for the original 5-min data) to 0.380 (for L1 interpolated 3-min data).

The effects of adding coloured noises to short data on the differentiation of RBR and the other breathing trials using Mann-Whitney tests were not encouraging. Nor was binarising the data (using Petrosian’s first three methods [17]). Resulting effect sizes were all <0.230.

### 3.3. Parameter Tuning

Results for the CEPS measures used can be found in the Appendix A.

### 3.4. CEPS, DynamicalSystems.jl and Other Analysis of RRi, Respiration and EDA Data

For the RRi data, results from DynamicalSystems.jl and CEPS were identical for PE with *m* = 4 and lag = 1. FD_H results differed for the RRi data tested, for two reasons: (1) In DynamicalSystems.jl, values of *k* to compute the Higuchi lengths *L*(*k*) were selected using logarithmically spaced values from 2 to about 2^7^, based on time series length. In CEPS, on the other hand, *k* values were from *k* = 1 to *k*_max_ = 2 to 15 (i.e., linear spaced values up to a varied *k*_max_, choosing the *k*_max_ with best discriminatory power for our application); (2) different line-fitting functions were used in DynamicalSystems.jl (fitting a slope to an identified linear scaling region as described in Datseris et al. 2021 [6]) and CEPS (standard MATLAB ‘polyfit’ polynomial curve fitting).

#### 3.4.1. Five-Minute ECG RRi Data—CEPS, DynamicalSystems.jl and Kubios HRV Analysis

Differences with BrPM were analysed for the various families of measures described above (FD, HRA, PE, RQA, OC and OE). Friedman’s *χ*^2^ and Kendall’s *W* were used, as described in Section 2.2.9). Medians are shown in Table 6, with interquartile ranges (IQRs) in parentheses.

Values of *χ*^2^ and *W* were greater for the permutation entropy family of measures than for the others, except for the RRi data resampled at 10 Hz, for which *χ*^2^ and *W* were greatest for the HRA family of measures.

For the 5-min RRI data resampled at 4 Hz, Kendall’s *W* was also computed for the additional measures from DynamicalSystems.jl but was only > 0.3 for wavelet entropy (‘wavent’); *W* was very small indeed (~0.02) for both Δ and Δ_2_, if slightly greater for the former.

Those measures from each family which performed best (i.e., with values of χ^2^ > 150) are shown in Table 7, with results for the best-performing 4R measures from Kubios HRV provided as a comparison (see [138] for details).

Of the CEPS measures, the FDs were most useful, for all data types, and outperformed all the Kubios HRV measures except PLFP (peak low frequency power), itself one of the measures used to define RBR, as explained above. In particular, FD_H performed well, as in our previous study [4]. The Kubios HRV frequency-domain measures based on the autoregressive (AR) method provided better differentiation between paced breathing rates than those based on the Lomb-Scargle periodogram (which, however, does not require equal resampling of the RRi data).

In addition to analysing RR interval results for those ‘top’ measures which performed best, they were also analysed for the CEPS and Kubios HRV measures that did *not* appear to be greatly affected by BrPM (i.e., with Friedman’s *χ*^2^ < 10). For the RRi data resampled at both 4 Hz and10 Hz, these were FD_Moisy_Box (FD), LZC (OC), SlopeEn (OE), and two RQA measures (RTmax and Lmax); for the data resampled at 4 Hz, they also included SI (HRA) and AAPE (PE), with GridEn (OE) for the data resampled at 10 Hz. For the un-resampled data, measures were EPP SD2_6 (HRA), CAFE (OE) and three RQA measures (including Lmax once again). Of the Kubios HRV measures, HFpwr (AR) also appeared to be little affected by respiration rate.

As illustrated in Figure 5 and Figure 6, the ‘top’ measures showed noticeably clear differences in measures with breathing frequency and RBR (PLFP), with two patterns predominating: (a) Increasing from Baseline to 5 BrPM, decreasing with higher BrPM, and finally increasing again at RBR (which was in any case 5 BrPM for most participants); (b) Decreasing from Baseline to 5 BrPM, increasing with higher BrPM, and finally decreasing again at RBR. In other words, compared with free breathing, fixed breathing rates (and ratios) increase the pattern ‘a’ measures, but decrease the pattern ‘b’ measures.

##### Post-Hoc Analysis

Conover tests were conducted and intraclass correlations computed for both non-resampled RRi data and for the same data resampled at 4 Hz.

More significant differences in pairs of trials were found with Baseline than with Self-paced breathing, with most such differences occurring between 5.0 or 5.5 and the other BrPM. Fewest significant differences were found between 7.0 and the other BrPM.

For the 5-min non-resampled (noR) data, 15 of the 153 CEPS measures analysed decreased between Baseline and RBR with standardised values of the Conover statistic ≥ 0.8 (including six PE-based, seven FDs, one Poincaré-derived), and seven increased (including one PE-related and six Poincaré-derived or HRA). For the resampled (4R) data, eight of the 159 CEPS measures decreased (including five FDs and two Poincaré-derived or HRA), but only two measures increased (including one Poincaré-derived). None of the six DynamicalSystems.jl (DS) measures tested showed values of the Conover statistic ≥ 0.8, although for DS wavelet entropy, the value was 11.787 (standardised value 0.762), increasing in 43 participants, and for DS FD_H it was again 11.787 (standardised value 0.762), increasing in 8 participants.

Four CEPS measures decreased between Baseline and RBR for both the noR and 4R RRi data. They were all FDs: FD_C, FD_H, mFD_M and FD_PRI. Only one measure (SDNNdown) increased for both the noR and 4R RRi data. Results were not dissimilar for the 10 R RRi data (Table 8), although Bubble entropy (BE) and MmSE at scale 10 also decreased significantly, and Robust CV (RoCV) increased (not shown in Table 8).

Somewhat more CEPS measures showed standardised ICC (Intraclass Correlation Coefficients) than Conover S values ≥ 0.8 (87 of 153 noR measures; 74 of 159 4R measures).

#### 3.4.2. Respiration Data—CEPS Analysis Only

Unsurprisingly, many more measures differentiated between breathing rates for the respiration (RSP) data than for the RRi data. Numbers of measures for which Friedman’s χ^2^ > 150 are shown in Table 9. Note that maximal χ^2^ values are lower than for the RRi data.

The ‘top’ measures showed noticeably clear differences in measures with breathing frequency and RBR (PLFP). Figure 7 shows the ‘Top’ measures, by family, with values of Friedman’s χ^2^. The measure MmSE13 is not included in this Figure, as although it was highest at baseline it remained unchanged for all paced breathing rates.

The ‘top’ FDs, PE-based measures, ESCHA_d and other entropies were all highest at Baseline and increased with respiration frequency. The opposite pattern was found for the EPP SD2 (HRA) measures, while CCM (OUT) neither increased nor decreased monotonically with breathing frequency. FD_PRI (Raw RSP) decreased less from Baseline during Self-paced breathing than the other FD measures (IN, OUT or PP).

Respiration data were only analysed using the Friedman and Conover test, not the ICC method. For the raw respiration data, FD_PRI was the only measure out of 102 analysed that resulted in a standardised value of Conover S ≥ 0.8 (better differentiation between baseline and RBR). For the respiration interval data, 20 of 196 INbreath measures, 23 of 196 OUTbreath measures and 36 of 195 breath peak-to-peak measures resulted in standardised S ≥ 0.8. Non-standardised values of S were more often maximal for the peak-to-peak (17 measures) than for those taken from the INbreath (4) or OUTbreath data (3). For all three data types, the measure that resulted in the highest value of *S* was MmSE at scale 13, followed by LLE at various iterations.

#### 3.4.3. EDA Data—CEPS and DynamicalSystems.jl Analysis

No measures differentiated between breathing rates for the EDA data with Friedman’s χ^2^ > 150. For the following measures, Friedman’s χ^2^ was greater than 28: RMSSD (χ^2^ = 29.035), EPP SD1_1 to SD1_-7 excluding SD1_6 (χ^2^ = 28.303–28.824), and FD_K (χ^2^ = 28.767).

In contrast to the RRi and respiration data, no measures increased or decreased consistently for 35 or more participants between baseline and the aggregate trial (RBR). Non-standardised values of Conover *S* were thus consistently low (Median 1.732, IQR 0.905 to 2.613). Measures with standardised values ≥ 0.8 for both Conover S and ICC were only SD1, SD1_1 to SD1_7 and RMSSD, which all showed 30 or more increases between baseline and RBR.

EDA in the first 23 cases (161 trials) examined tended to decrease during each five-minute recording. For the whole cohort, CEPS measures of Robust slope (RoSlope) were thus predominantly negative as would be expected when sitting quietly. This was the case for 35 or more participants when breathing at 5 or 6.5 BrPM, and for 37 participants during RBR.

#### 3.4.4. Summary of Results for RRi, Respiration and EDA Data

Table 10 and Table 11 summarise the top Friedman test results for differences in CEPS and DynamicalSystems.jl measures between all eight trials, for the RRi, RSP and EDA data, with corresponding results for the Kubios HRV measures.

Note that, for the same data, *χ*^2^ and *W* are lower for the top two DynamicalSystems.jl measures than for the corresponding CEPS measures. Only wavelet entropy (Wavent) shows a reasonable effect size (Kendall’s *W* > 0.4). Patterns of change with breathing frequency differ from those observed for the corresponding CEPS measures (compare Figure 5, Figure 6 and Figure 7 above with Figure 8 below).

The following Table 12, Table 13 and Table 14 present a summary of which measures resulted in the ‘top five’ non-standardised values of Conover S for the different data types, for all 28 pairs of trials rather than the Baseline-5 BrPM pair only.

Note that the highest median values occur for the non-resampled RRi data, followed by the INbreath interval data. Lowest values occur for the EDA data.

At the other end of the spectrum. no differences in Conover S were observed for a number of measures and paired Trials, for each of the various data types.

For RRi data, for example, measures which appeared less affected by respiration rate included some of the RQA, Jitter (frequency variation from cycle to cycle [101]), LLE, EPP r measures and LZC. For the other data types, different groupings of measures were unresponsive to respiration rate, but there was no obvious pattern to these.

#### 3.4.5. Some Findings on Heart Rate Asymmetry (HRA)

As mentioned above (Section 2.1.1), measures of heart rate asymmetry (HRA) have been used to quantify differences between phases of acceleration and deceleration in heart rate data. Given the intimate relationship between HRV and respiratory sinus arrhythmia (RSA) [73], we expected to find that an individual’s RBR would be reflected in their HRA.

The five ‘classical’ HRA measures and 11 derived from Poincaré plots behaved in interesting ways. Whereas the median values of Ehlers’ Index (a linear measure) were maximal at 6 BrPM and minimal during Self-paced breathing, the other classical HRA measures and Rohila’s ASI demonstrated a variety of changes with respiration frequency (Figure 9).

Figure 9 demonstrates that the HRA measures, in general, differed between Baseline, Paced and RBR trials.

Post-hoc analysis for the paired RRi noR data Baseline-RBR trials using Conover’s *S* indicated that SDNN_up_ and SDNN_down_ performed best (both with *S* > 12), followed by SD2_up_ (*S* = 11.982). Values of *S* > 10 were also obtained for PI, GI and SD1down. By way of comparison, FD measures such as mFD_M, FD_PRI and FD_H all resulted in values of *S* > 14. For 10 out of the 16 HRA measures analysed here, Conover’s *S* was lower for the RRi data resampled at 4 Hz.

##### Correlations between HRA Indices and HRV Measures

In our literature review on HRA (Section 2.1.1), we cited one study questioning “the widespread belief that it is the parasympathetic branch of the autonomic system which is responsible for decelerations and the sympathetic branch which is responsible for accelerations” [31]. We therefore explored correlations between the 16 HRA indices (and the Complex Correlation Measure, CCM [34]) and 13 of the standard Kubios HRV measures that have sometimes been considered to reflect parasympathetic or sympathetic activation [139,140]. To keep the analysis manageable, we considered only ‘strong’ correlations, with Spearman’s |*rho*| > 0.9.

With minor variations, there was a clear pattern of results that was very similar across all seven trials, as well as the composite RBR trial. Strong correlations occurred between only five or six of the HRA indices (SD1up/down, SD2up/down and SDNNup/down), and seven or eight of the HRV measures (SDNN, RMSSD, HF or LF power or log power, Total power and the Kubios ‘PNS’ measure). Negative correlations with the HRA indices were all with Baevsky’s ‘Stress index’ of sympathetic activation [141]. Positive correlations between the HRA indices and HF power or log power occurred in all trials, but with LF power or log power only in the externally paced trials. Strong correlations occurred slightly more often with the HRA indices calculated for the RRi data resampled at 4 Hz than for the non-resampled data, but values of |*rho*| were not consistently larger for either data type across all trials. Overall, excluding the composite RBR trial, most strong correlations were found for the HRA indices SDNNdown (39) and SD1down (37), and for the HRV measures SDNN (39) and Total power (35). For correlations between the PNS measure and the six HRA indices, *rho* was consistently > 0.6 in all trials, for both the ‘up’ and ‘down’ version of these indices. Correlations were weaker for the classical or normalised HRA indices, ASI and CCM. However tantalising these findings, there was thus no obvious pattern that would indicate an association between parasympathetic or sympathetic activation and any particular HRA measure.

A corresponding analysis of noR and 4R RRi data was carried out for the families of 22 FD measures and eight PE-based measures. There were no strong correlations, either positive or negative, between the PE-based and HRV measures.

For the fractal dimensions, Spearman’ |*rho|* was > 0.9 mostly for FD_K (14 negative correlations, 97 positive). Correlations were again negative for Baevsky’s Stress index (for both non-resampled data and data resampled at 4 Hz). For both data types, positive correlations were with Kubios HRV’s PNS index, LF *and* HF power and log power, Total power, SDNN and RMSSD. The correlation between FD_K and PNS for the 4R data was slightly less strong than the others (*rho* = 0.835). There were also three strong correlations for FD_H (two negative, with DFA *alpha*1, one positive, with HF relative power), one negative for mFD_M (with DFA alpha1), and two for NLDw mean variants, again with HF relative power.

One possible interpretation could be that FD_K, FD_H and the NLDw variants are more associated with parasympathetic than sympathetic activation.

Results for the HRA and FD measures were then compared. Excluding the composite RBR trial, 3332 correlations between HRA indices and HRV measures were analysed for the remaining seven trials. Of these, *rho* was <−0.9 for 53 (1.59%), and >0.9 for 344 (10.32%). For the FDs, there were 4312 correlations in total, for 15 (0.35%) of which *rho* was <−0.9, while for 85 (1.97%) *rho* was >0.9. However, median values of *rho* for the correlations with rho < −0.9 or >0.9 were larger for the FD than HRA measures: −0.936 and 0.961 for the FDs (IQRs –0.944 to –0.928 and 0.929 to 0.973, respectively), but only –0.931 and 0.934 for the HRAs (IQRs –0.948 to –0.918 and 0.915 to 0.968, respectively). Thus, although there were numerically more strong correlations with HRV measures for the HRA indices than for the FDs, these correlations were not necessarily stronger.

##### Respiration and Asymmetry

For the 284 respiration recordings, the median Outbreath-to-Inbreath ratio (RespR) was 1.424 (IQR 1.239 to 1.650). Further results are reported in the Appendix A.

#### 3.4.6. Difference and Agreement between Baseline or Self-Paced Breathing and Optimal (or ‘Resonance’) Breathing or Breathing at 5 BrPM: Do Measure Values during Slow Self-Paced Breathing Predict Those of RBR?

As described above (Section 2.2.1), resonance breathing can be defined in a number of different ways. In this paper, we used the latter definition of RBR as peak low frequency power (PLFP).

For PLFP, 5 BrPM was found to be the RBR for 15 participants (34%), 5.5 BrPM for 12 participants (27%), 6 BrPM for 9 (20%), and 6.5 BrPM for 6 participants (14%) and no one at 7 BrPM. For the remaining two participants, PLFP was lower during externally paced than during slow self-paced breathing, with rates of 4 and 8 BrPM. Similar results were obtained for RBR defined as LFBP. No participants showed an RBR of 7 BrPM, with either definition. For 39 participants, the two definitions resulted in the same RBR.

Table 15 below shows median values of Conover’s *S* for differences between Self-paced breathing and RBR, between Baseline and RBR, between Self-paced breathing and 5 BrPM, and between Baseline and 5 BrPM, for all measures tested, for the various data types. Numbers in parentheses indicate the total number of measures analysed for each data type.

The post-hoc analysis of variance thus indicates, for these four comparisons, that for the RRi data, there was least difference between self-paced breathing and breathing at 5 BrPM, whereas for the EDA and respiration data (whether raw of interval), least differences were between self-paced breathing and RBR (lowest median values for each data type in bold).

Spearman’s *rho* was computed for 324 comparisons of RBR Slots (using the PLFP definition) with baseline, and for 324 comparisons of RBR Slots with Self-paced breathing. Of these, 98 showed *p* < 0.001. These were all for RR interval data; *none* of the correlations calculated for the Respiration data (whether IN, OUT, OUT/IN Ratio or Peak-to-Peak) achieved that level of significance.

HRV measures which predicted RBR for RRi data with *p* < 0.001 and *rho* > 0.6 were tabulated (45 measures at Baseline and 36 during Self-paced breathing). NO frequency domain measures showed *rho* > 0.8, and of the usual HRV measures only Mean RRi and Mean HR showed *rho* > 0.9.

Early analysis suggested no correlation between Baseline or Self-paced breathing rates and RBR, with neither appearing to predict RBR.

However, for the respiration data, several CEPS respiration measures predicted RBR with *p* < 0.01 (but ≥ 0.001), 0.396 < *rho* < 0.508, particularly for INbreath and OUT/IN Ratio data. For the former, FD_K, FD_M, EPE, CCM and two HRA indices (SI and AI) appeared useful; for the latter, FD_S, T_E Tone and 6 measures derived from the Poincaré Plot.

Note that for the respiration data, Self-paced breathing results provided better predictions than Baseline breathing, whereas results were similar for both for the RRi data (slightly more predictions using the Baseline data). *Correlations* between measures in the same paired trials for which Spearman’s *rho* > 0.7 for all four comparisons were then examined. Results can be found in the Appendix A.

#### 3.4.7. Results for Correlations within ‘Families’ of Measures, and between Individual Measures When Applied to Different Data Types (RRi, Respiration and EDA) Are Described in the Appendix A

### 3.5. The Effects of Time

#### 3.5.1. Data Length and Its Effect on Different Measures

Whereas linear measures such as the median or (robust) coefficient of variation can be calculated using very few data points, most nonlinear measures require much longer data for their estimation. In an initial test, we found that some, such as EPE, may be calculated for as few as 10 points (RR intervals), and that PJSC, RPE and TPE (Tsallis Permutation Entropy) can be calculated for 15 data points, with relatively stable values (CVs) over longer segments, but it was still not clear how meaningful results would be for such short data. On the other hand, DistEn is known to provide acceptable results for as few as 50 data points [142].

Strictly speaking, there will be complete agreement between values of the same measure computed for different data lengths only if the data exhibits at least ‘weak-sense’ stationarity (see [4] and the online CEPS ‘Primer’ [https://github.com/harikalakandel/CEPSv2/tree/master/doc] (accessed on 20 January 2023) for a list of measures which require this, and for further information). Three simple tests for non-stationarity by Etienne Cheynet are implemented in CEPS [https://github.com/harikalakandel/CEPSv2/tree/master/] (accessed on 20 January 2023). Here we used the reverse arrangement test for 1-, 2-, 3-, 4- and 5-min non-resampled RRi data segments from 326 recordings (all five starting at the same point in time). For these recordings, only 28 showed no non-stationarity for any of the five data segments, while 70 were non-stationary for one segment, reducing to 41 being nonstationary for all five data lengths. Non-stationarity for all five segments was lowest at Baseline, and highest for the composite RBR trial.

Caution should thus be observed when interpretating results for measures which—at least in theory—require stationarity of data, or which have not been thoroughly tested on short data.

##### Data Length and Differences in Measures between Breathing Rates

Having conducted Friedman and Conover tests to explore differences between trials for the various data types used, we then examined how RR interval data length affected Friedman test results when differentiating between several FD measures for the different paced breathing rates. Using 1-, 2-, 3-, 4- and 5-min data segments of non-resampled data for instance, we noted that differences between trials for the Katz, Mandebrot or Sevcik FD measures were not significant (*p* < 0.001) even for the full 5-min recordings we used (although FD_M and FD_S results were significant for 1-min data), whereas others, such as the box-counting algorithm of Meerwijk and van der Linden, provided significant results for 3- and 5-min data, but not 4-min data. Only Castiglioni and Higuchi FD, and Maragos’ multiscale FD, showed significant results for all five data lengths, with the latter providing the largest values for Friedman’s *χ*^2^ and Kendall’s *W* at 5 min, but FD_C and FD_H providing more consistently similar values across all data lengths (median *W* 0.371, IQR 0.354 to 0.720, and median *W* 0.632, IQR 0.599 to 0.633, respectively).

Three data types were analysed for the effects of data length: non-resampled RRi, RRi resampled at 4 Hz, and EDA. As a comparison, the HRV measures in RR-APET were also analysed, median values of *χ*^2^, Kendall’s *W* and the Conover statistic for the Baseline to RBR differences were calculated for each measure in the three main families (FDs, HRAs and PE-based measures). Measures were considered if they showed standardised median values of Friedman’s *χ*^2^ and Kendall’s *W* > 0.8, or of the Conover statistic *S* for differences between Baseline and the composite RBR trial. Numerically, these thresholds were approximately 150, 0.480 and 12, respectively, for the non-resampled RRi data, 145, 0.466 and 10.5 for the data resampled at 4 Hz, and 20, 0.070 and 3.3 for the EDA data. The corresponding thresholds for RR-APET were 100, 0.319 and 9.4. The EDA thresholds were low, unlikely to be useful in practice, so were not examined further here, nor were any measures with values lower than any one of these thresholds for one or more of the 2-, 3-, 4- or 5-min data segments. This draconian limitation reduced the number of measures that might be serviceable for analysis of short data *and* for differentiation between trials to something manageable, although excluding many measures that might otherwise have been useful for one *or* the other, particularly for the RRi (4R) data. Results are shown in Table 16.

Conducting Friedman tests and comparing medians are of course sensible strategies to use to determine group effects. However, were all differences in a measure between trials (e.g., Baseline to RBR) in the same direction for all study participants and all durations? (In a preliminary examination of the data, for example, it was found that one measure, multifractal FD (mFD_M), differentiated well between paced breathing at 7 BrPM and the other paced breathing rates for all data lengths, but that the sign of the difference flipped between the two shorter and three longer lengths.)

Negative and positive differences in 220 CEPS measures between Baseline and the composite RBR trial were counted, for 1-, 2-, 3-, 4- and 5-min data, for both the noR and 4R data, and also for 22 HRV measures output by RR-APET (excluding those for ‘ultra-low frequency’). Differences were calculated case by case, not between group medians.

For none of these measures were there 220 positive or negative differences for all five durations (220 = number of durations * number of participants), mostly because of zero differences between Baseline and RBR for some participants. If the 1-min data were excluded from the analysis, still no measure exhibited 176 (4 × 44) differences in the same direction. The ‘top 13′ measures for each RRi data type (i.e., with most differences of the same sign across the 1-, 2-, 3-, 4- and 5-min data, or excluding the 1-min data) are listed in Table 17, together with the ‘top 4′ HRV indices.

A more formal reliability analysis was then conducted, using the ‘intraclass correlation coefficients’ (ICC) method as described above, with ICC values standardised between 0 and 1. Measures for which standardised ICC were >0.9 are in bold type in Table 17. In total, numbers of measures with ICC > 0.9 for the different data types were as follows: 26 (28.6%) for EDA, 53 (34.9%) for the non-resampled RRi data, 92 (25.2%) for the RRi data resampled at 4 Hz, and 6 (27.3%) of the RR-APET measures. The ICC method thus appears to be less stringent than the less formal top-slicing method. Incidentally, there were no obvious patterns for those ‘contrarian’ participants where differences were in the opposite direction to everyone else’s, whether in terms of questionnaire scores, age, sex or the usual Kubios HRV measures.

Note that five of the ‘top 13′ measures for the non-resampled data in the above Table are permutation-entropy based, while three of those for the resampled data are FDs, four are largest Lyapunov exponents at different scales, and two are for RCmDE at different scales. RoCV is the only measure that occurs for both noR and 4R RRi data. Highest counts were for the RR-APET measures, though for only one of these, the arguably nonlinear measure SD2, was the ICC > 0.9.

##### Agreements between Measures for Different Data Lengths

Findings using Spearman’s *rho*, the ICC and percentage differences between measures for the segmented and 5-min data were then compared. Some results are shown here for the Baseline and RBR Slots. Only the 38 measures included in Table 16 and Table 17 were considered—21 for the RRi noR data, 13 for the RRi 4R data, and 4 HRV measures from RR-APET.

Figure 10 shows how correlations increase, as expected, with segment duration, for all three sets of measures. Note that the patterns of correlation are different for the three sets, with lowest values of *rho* during Self-paced breathing for the RRi (4R) measures, but when breathing at 5 BrPM for the RRi (noR) and RR-APET measures.

Percentage differences between median values of the measures for the segmented and 5-min data were also revealing (Figure 11).

As shown in the top row of Figure 11, differences between AE and FD_H for short-duration and 5-min data were greater for the RRi (4R data than for the RRi (noR) data. In the middle row, it appears that patterns of percentage difference were quite different at Baseline and during breathing at 5 BrPM. Insets show coefficients of variation (CVs) for the different measures, and in the bottom row, FD_PRI and CPEI showed less variation at Baseline than both the RR-APET measures, although this was no longer the case for FD_PRI during the composite RBR trial.

The three methods used here (ICCs, Spearman correlations and percentage differences between measures from the 5-min and shorter data) provide complementary information, and all appear to be useful. As might be expected, median CVs for all 38 measures and all 8 trials taken together (i.e., including the RBR trial) were strongly (and negatively) correlated with the number of times values of the measures for the 1-, 2-, 3- and 4-min data were within ± 5% of their values for the 5-min data (Pearson’s *R* = 0.952). In contrast, those counts were not significantly correlated with the ICCs.

Measures with short-data values consistently within ± 5% of their values for the 5-min data, for 31 or 32 of all the 32 duration * trial combinations are listed in Table 18, together with the five measures with short-data values least often within ± 5% of their values for the 5-min data.

Five of the top 12 measures are FDs and 1 is PE-based; none are HRAs.

Figure 12 summarises results for the measures considered so far, based on standardised values of Conover *S* and ICCs. Points on the diagonal lines represent measures with equivalent performance in both tests (Conover and ICC); those above the line correspond to measures with higher ICC, and those to the right of the line to measures with higher *S*. Points within the rectangular boxes at the top right of each scatter plot show measures which performed best in both analyses.

Note that S and ICC were >0.9 for eight RRi (noR) measures, including three FDs, with *S* and ICC > 0.8 for a further 10 measures. Highest *S* (>0.999) was for FD_H, and highest ICC (>0.99) for EPP SD1_3. In contrast, no RR-APET HRV measures resulted in *S* and ICC > 0.9, although this was the case for one of the RRi (4R) measures (AE), and for three based on EDA data (RMSSD, and EPP SD1 at lags 1 and 2). However, *S* and ICC were >0.8 for SD2 (RR-APET), and for a further five RRi (4R) measures (FD_H, mFD_M, EPP SD2_5, SD2_down_ and SDNN_down_), with wavelet entropy providing the best ICC (0.991) of the RRi (4R) measures, although a low *S* (0.762), and FD_PRI providing a high *S* (0.966) but a low I*CC* (0.599). For the EDA data, ICC was >0.999 for ‘Jitta’ and 0.966 for FD_K, although *S* was not > 0.8 for either of these measures.

#### 3.5.2. Do Nonlinear Measures Indicate RBR More Accurately than Standard HRV Measures, Especially for Short Data?

Changes with data length at Baseline and during the composite RBR trial are shown for selected nonlinear (FD, PE-based and OE) measures and for two linear RR-APET measures above, in Figure 11. Here, using the effect sizes (Z/√n) from Mann-Whitney tests with significant results (*p* < 0.001) for four or more of the 1- to 5-min durations, we focus on those FD, PE-based and HRA measures which best differentiated between RBR and other Paced breathing rates. Results were different for the RRi (noR) and RRi (4R) data.

For the RRi (noR) data, only FD_C, FD_H, NLDwL_m and NLDwP_m out of eight FD measures showed effect size > 0.3 for any durations, although effect size for seven of these was consistently >0.25 for at least four of the durations. mFD_M, the remaining FD measure, was >0.2 for all durations. Of seven PE-based measures, all except RPE achieved an effect size > 0.25 for four durations, with effect size for EPE and ImPE being > 0.3 for the 4- and 5-min data. No effect sizes for any HRA measures even approached 0.25, although both PI and C1_a_ achieved effect sizes > 0.2.

For the RRi (4R) data, only FD_C and FD_PRI showed effect size > 0.3 for any durations, with only mFD_M of the other FD measures having an effect size > 0.25 for three durations. Of the seven PE-based measures, only CPEI achieved an effect size > 0.25 and mPE_M an effect size > 0.2, both for four durations; for no measure was effect size > 0.3, and for all PE-based measures, effect size was consistently lower for the RRi (4R) than for the RRi (noR) data. Of the HRA measures, both PI and C1_a_ achieved effect sizes > 0.2, as for the RRi (noR) data.

Of the RR-APET HRV measures, only the nonlinear measure DFA Alpha2 achieved an effect size > 0.3, while that for Alpha1 did not reach 0.25. The most stable measures over the five durations were SDNN and LF peak frequency, but effect sizes for these did not even reach 0.15. Effect sizes for the nonlinear RQA measures DET and L_max_ were sometimes >0.25, but were also very variable over the different durations, while for LF power the effect size only exceeded 0.25 for the 5-min data.

Some of these findings are illustrated in Figure 13, together with maximum and median effect sizes, interquartile ranges and CVs for the best-performing measures for the different data types. Clearly, whether data is resampled or not is an important issue in this context.

It can be seen that RRi (noR) effect size (ES) is rather constant for FD_H and FD_P from 2–5 min. but that for FD_C it increases with data length and is around 0.3 (0.296 or greater) from 3 min and upwards. FD_PRI varies greatly with duration for both the RRi (noR) and (4R) data.

Of the RR-APET measures, only DFA Alpha2 attains an ES of 0.3. ES for LF peak frequency varies less with data length than that for LF power, but lower CVs were obtained for the three FDs, especially FD-H.

In summary (and taking Table 18 into account), it would appear that FD_H, the NLDwL_m and NLDwP_m measures and FD_PRI offer good stability over durations of 3 min or longer as well as reasonable effect sizes for differentiating between RBR and the other breathing rates (including Baseline and Self-paced) when using RRi (noR) data. EPE and ImPE may also be useful for the non-resampled data. For data resampled at 4 Hz, FD_PRI and FD_C may be useful, with slightly greater effect size but marginally less stability than for the RRi (noR) data; the PE-based and HRA measures are less useful.

## 4. Discussion

Challenges and difficulties.

### 4.1. General Points

CEPS is a continuously evolving project. One of the primary objectives here was to compare findings when using a variety of CEPS fractal dimensions, Heart Rate Asymmetry measures, and others based on permutation entropy (see list of objectives). We investigated whether there were marked differences between the effects of paced, self-paced and non-paced breathing on a variety of physiological data (as in our first published CEPS paper), for example, which measures are most/least responsive to changes in breathing rate. We also wanted to compare our results using CEPS with those obtained using DynamicalSystems.jl, especially for FDs.

It has been particularly challenging when working with multiple datasets and measures to know how best to steer a clear course across the resulting ocean of results. There is always a risk of becoming either too restrictive (in essence barely extending or even merely repeating what has been done before) in order to maintain scientific credibility, or, at the other extreme, trying to include too much and losing one’s way. As authors, we all had different objectives for this paper, but hoped that by reporting only our main findings in the paper itself, and relegating subsidiary and subgroup analyses to the Appendix A, we have managed to keep the main message clear while also presenting a number of other potentially useful or important findings without too much self-indulgence or wasting our limited human and machine time resources.

After some initial exploration, we focused our attention on three families of measures—fractal dimensions (FDs), heart rate asymmetry measures (HRAs) and those based on permutation entropy (PE). Counts were made of how many times the abbreviation for each CEPS measure appears in this paper (usually with significant or otherwise meaningful findings). Those measures that appear most often in the paper are indeed, and inevitably, from those families—FDs 71, HRAs 70, PE-based 40 (with all OC and RQA measures occurring 14 times each, and OE measures only 11 times), Individual measures that appeared more than 10 times were mostly FDs—FD_H (40 occurrences), FD_PRI (37), mFD_M, and the various NLD subtypes (both 24) and FD_C (18). Of the HRA or Poincaré-based measures, EPP at various lags occurs 27 times, and of the OC family members, LLE at various lags appears 20 times. From the PE-based family, CPEI is present 14 times, and mPM_E 12 times. Only one member of the OE family occurs more than 10 times—MmSE, a variant of SampEn suited to short data. Other conditional entropies (e.g., ApEn, FE and SampEn itself) generally require longer data and occur only infrequently. Correlations with HRV measures could not be easily interpreted in terms of autonomic activation. A subsidiary challenge has been in managing both the large amounts of data collected and also the codebase itself. We have certainly experienced problems with code that does not do what it is supposed to, seems to break over time and is ‘buggy’ [143]. To help avoid such problems, users of CEPS are encouraged to report the problems they encounter. Another challenge: after working in relative academic isolation for several months during lockdown, we had to deal with rigorous questions from George Datseris, our new co-author, on the appropriateness of applying FD measures willy-nilly to non-equally sampled data. As a result, we had to rethink our whole approach. This has made, we hope, for a stronger paper, but also made for a lot of unanticipated work. For DynamicalSystems.jl measures, in particular FD_H, equal resampling is generally considered necessary. On the other hand, we discovered that none of the six DynamicalSystems.jl (DS) measures tested showed values of the Conover statistic ≥ 0.8, and indeed, overall CEPS resulted in larger values of S for Conover tests than did DynamicalSystems.jl, for the data and comparisons we used. With CEPS, we found that pragmatically useful results can be obtained even for data that are not equally sampled, although results for equally and non-equally sampled data will be different. In particular, resampling was found to make a considerable difference to numbers of significant correlations between measures, and thus it is important to consider carefully whether or not to resample data prior to using CEPS. Another issue is the non-stationarity of data. In theory, measures such as PE and wavent are suitable for use with non-stationary data, whereas many others are not. As an example, not all researchers agree that FD_H is appropriate to this situation [144]. Here, however, we found that FD_H, as well as other FDs and PE-based measures, provided useful results despite data non-stationarity.

As a small research group with other professional responsibilities and limited expertise in some areas, developing CEPS in addition to our other commitments has at times been very demanding. A major challenge, and one that still delays implementation in CEPS of several measures, has been the translation of code into MATLAB from other programming languages, particularly when language-specific libraries are involved (sometimes known as ‘dependency hell’). However, the ideal mix of skills within the research group meant that these restrictions, and others, were eventually overcome.

### 4.2. Our Basic Approach

There is a problem with our approach in that in a sense it involves a circular argument: If RBR is defined with reference to standard HRV measures such as PLFP, how can any other measures, whether nonlinear or not, be considered as providing better differentiation of PLFP (or LFBP, for that matter) from Baseline or Self-paced breathing? We believe that the answer to that lies in the statistical methods used: If Friedman tests (*χ*^2^, with Kendall’s *W* as a measure of effect size) and Conover tests show more significant differences for the nonlinear than the standard HRV measures, and if these differences also hold for shortened data, then further use of carefully selected nonlinear measures from those available in CEPS can be justified. However, caution should still be exercised when interpreting results from the Conover tests for non-resampled data, in that variance may be greater than when data have not been equally resampled. Pragmatically, we have found FD_H to be a useful measure for differentiating the effects of breathing rate on non-resampled RRi data. On the other hand, FD_H, as well as several other FD measures, did not result in useful values of Friedman’s *χ*^2^ for either 5-min or shorter RRi (4R) data, whereas they did for the non-resampled RRi data, and the clear differences for the two data types in Figure 13 reinforce the importance of careful selection of the type of data to use and careful interpretation of results. For RRi data, for example, measures which appeared less affected by respiration rate included some of the RQA, Jitter (frequency variation from cycle to cycle [101]), LLE, EPP r measures and LZC. For the other data types, different groupings of measures were unresponsive to respiration rate, but there was no obvious pattern to these.

### 4.3. The Anxieties of Data Collection and Collaboration

Data were gathered in 2021–22, during the COVID-19 pandemic, at a time of great collective anxiety including maintaining strict adherence to Governmental and institutional COVID-19 protocols. This may have impacted the state of mind of both participants and researchers. It certainly led to challenges in scheduling sessions and in requirements for conducting them safely. We also found how difficult it can be to work together in sporadic online collaboration and meetings. Our solution was to have more regular meetings using a centralised approach to meeting management, data recording, data sharing and a cooperative approach to data analysis and writing. Our ability to engage with research participants during the pandemic relied on persuading the ethics review board to accept our COVID-19 mitigation procedures based on risk vs benefit of the experiments which included addressing the possibility of increased risks if the pandemic was to get any worse.

### 4.4. Including EDA Results

EDA in the first 23 cases (161 trials) examined tended to decrease overall during each five-minute recording and for the whole cohort, CEPS measures of Robust Slope (RoSlope) were predominantly negative as would be expected when sitting quietly. EDA data from participants demonstrating frequent *increases* over time, or with greater EDA reactivity (to unexpected noises, for instance), could be explored for associations with age, sex or questionnaire characteristics, as well as with HRV, PRV and respiration measures and their changes. EDA is conventionally interpreted as an indication of psychological or physiological arousal and is known to reflect sympathetic activation. It would be of interest to investigate whether smoother descents indicate greater ability to be mindful or relax, and whether more erratic descents suggest greater stress.

Correlation analysis indicated that agreement for the RRi and EDA measures was *less* between Self-paced and RBR breathing than between the other paired trials analysed, but that agreement between *some* of the raw Respiration and RSP interval measures was indeed greater for the self-paced/RBR pair than for the other trial pairs. However, counts of pairs with numbers of measures having *rho* > 0.7 did not support the hypothesis that self-paced breathing predicts RBR, for the EDA data particularly.

### 4.5. An Explanation of HRA Results

The correlations of HRA with HRV measures for the RRi data could not be easily interpreted in terms of parasympathetic or sympathetic nervous activation. Reluctantly, we were forced to conclude, with [31], that Procrustean binary thinking cannot possibly do justice to the complexity of any associations that do exist between the divisions of the autonomic nervous system activation and heart rate accelerations or decelerations. It is, however, always tempting to associate particular measures with autonomic nervous system activation, whether these are HRAs or FDs. For correlations between the PNS measure and the six HRA indices, *rho* was consistently >0.6 in all trials, for both the ‘up’ and ‘down’ version of these indices. Correlations were weaker for the classical or normalised HRA indices and, however tantalising these findings, there was thus no obvious pattern that would indicate an association between autonomic NS activation and any particular HRA measure.

### 4.6. Limitations

We did not explore using very short data (<1-min), as we only became aware of the 2015 paper by Munoz et al. [75] on the validity of (ultra-)short recordings for HRV measurements when analysis of our own data was already well under way. Given that so many variables are involved, it might have been preferable to use a multivariate version of the Friedman test, such as the Friedman-Rafsky test [145]. A more formal regression analysis could have been used to assess whether and how well measures at Baseline or during Self-paced breathing predict RBR. Further limitations include the lack of easily useable frequency domain measures, the analysis and classifications sections of CEPS are not yet functioning, and the software is only useable with 1D data. In this respect EntropyHub has an advantage over CEPS at least for the latter limitation. CEPS is restricted in its repertoire of imported files, currently only accepting .txt, .mat, .csv and .xlsx. Although this appears to be a limitation, most data can be converted into one of the compatible import formats.

### 4.7. Advantages

Occasional feedback from users suggests that CEPS is reasonable easy to use. Using MATLAB App Designer, two new tabs have been introduced, for pre-processing and data modification. The first of these includes filtering, detrending and outlier removal options, and the second includes Normalisation (three methods), Increase (Interpolation), Reduce (i.e., coarse-graining), Add Noise (both noise and data-plus-noise files can be saved), Cut files (with or without overlap, etc.), Binarisation (three methods) and equal resampling (various methods available). The most significant advantage of CEPS is that it includes innovative complexity and an entropy measure mostly re-engineered from published literature or from personal communication with the authors. Experimental data shared between many of the external contributors ensured quality control over the analysis and inclusion of new measures.

In CEPS 2, as in CEPS 1, one useful feature is the ability to test the effect of parameter variation on the different measures and to visualise how multi-scaling may affect them. Another is an easy comparison of changes in different CEPS measures for the same or parallel physiological data streams. CEPS can thus be used to investigate the effects of paced breathing on ECG, PPG and RSP variability.

## 5. Conclusions and Future Directions

### 5.1. Conclusions

We have developed CEPS software as a physiological data visualiser enabling integration of state-of-the-art techniques. However, we are aware that CEPS is not unique, and that some biomedical scientists, in particular physiologists and clinicians, may find other packages more suited to their needs. We have designed the interface for clinical research with a structure designed for integrating new tools where users can switch seamlessly between tasks. For this project the aim was to strengthen collaboration between clinicians and the biomedical community, as demonstrated here by using CEPS 2 to analyse various physiological responses to paced breathing.

We observed that, while equal resampling is theoretically important for FD estimation, FDs provided pragmatically useful results even for data that was non-stationary and not equally sampled (although results for equally and non-equally sampled data were different). Given the number of times LLE and MmSE appear in this paper—despite not being members of the three main ’families’ of measures on which we focused—more attention should be paid to them in future research on resonance breathing.

CEPS 2 has a broad appeal as it is a cross-platform (Windows, Mac or Linux) MATLAB GUI which has proven to be more intuitive than command-line or menu-driven interfaces that rely on programming skills, as well as having a plethora of new tools. The program allows direct manipulation of graphical icons such as buttons, scroll bars, windows, tabs, menus, and cursors and allows the exchange of data between different software applications or data sets. Finding the correct measure is essential, however, and complexity and entropy changes often show more significant changes than conventional linear measures, particularly during paced compared to spontaneous breathing. As a conclusion, in its present form CEPS 2 is ideally placed to analyse respiratory-related data.

### 5.2. Future Directions

Measures newly implemented in CEPS 2, or in course of implementation, are asterisked in Table 4. Those measures planned for future inclusion are listed in parentheses (for measures already included in CEPS, see [4]). Please note that, although every effort has been made to implement these measures correctly in CEPS, time has not always allowed us to validate the results obtained when using CEPS with those researchers who provided us with code.

Future developments of CEPS will include a ‘plug-in’ facility to allow other researchers to add measures not already included in the list available, and Classification will be added as a further item in the Application Mode drop-down list, providing alternative methods of classifying results from using CEPS or from other sources. These are still planned. Ideally, an automated method of parameter tuning could be developed for use in different contexts, in order to speed up the process. This is currently beyond the scope of CEPS but is a future possibility.

Another avenue for future exploration would be to reduce data length even further than we did and average the values of appropriately selected CEPS measures over several ultra-short data segments—maybe even 10 or 20 s [75], rather than the 1-min minimum used in this study.

A further major extension to CEPS, as suggested by one of our reviewers, would be to include bi- or multivariate measures of complexity in the package, in order to permit fuller analysis of multichannel physiological recordings. Realistically, this may be beyond what our small research team can achieve, but we are always open to collaboration with others who have skills that may complement our own.

## Figures and Tables

**Figure 1 entropy-25-00301-f001:**
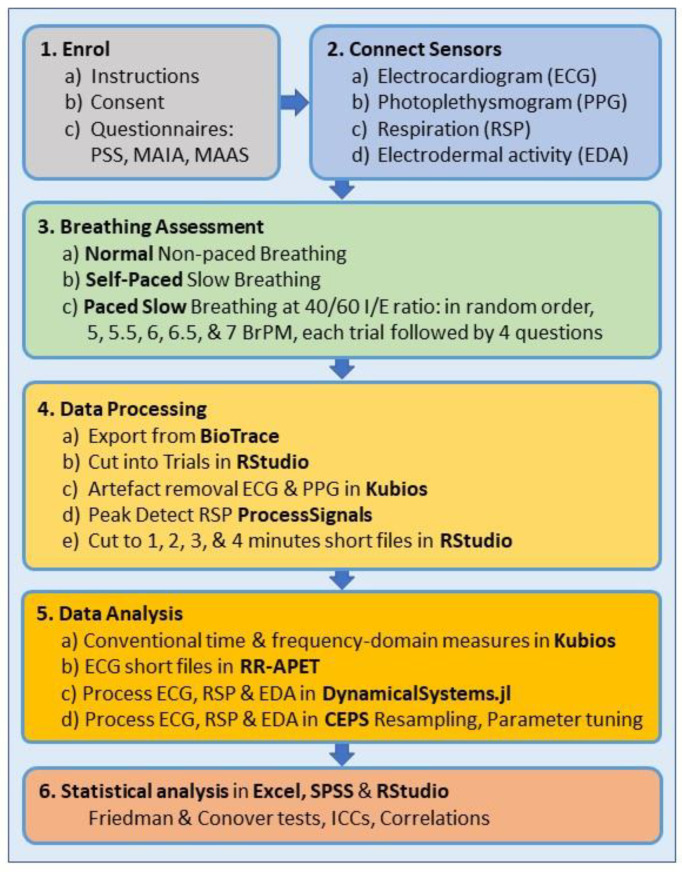
Outline of the study protocol. See list of abbreviations for interpretation.

**Figure 2 entropy-25-00301-f002:**
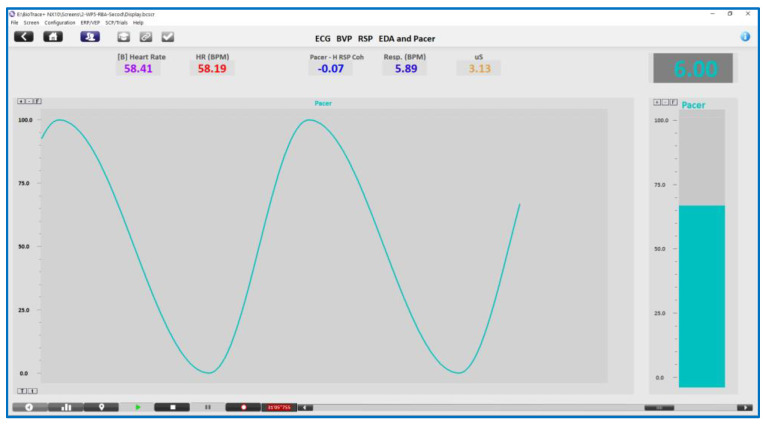
Example of the participants’ pacer display for a slow-paced breathing trial. The blue line on the left and the bar graph on the right rise and fall at a rate of 6 BrPM, i.e., 10 s per cycle with an inhalation/exhalation ratio of 40/60, or 4 s in and 6 s out, with no pause in breathing.

**Figure 3 entropy-25-00301-f003:**
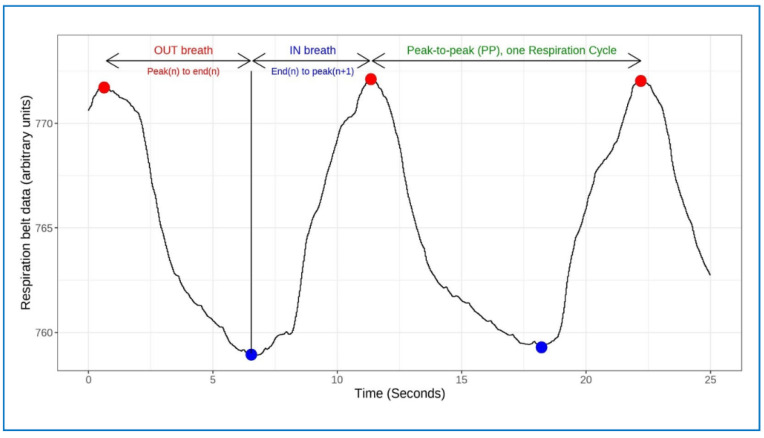
Respiration cycles at a 40/60 inhalation/exhalation ratio, showing INbreath (Blue to Red dot), OUTbreath (Red to Blue dot) and peak-to-peak (PP) (Red to Red dot) respiration intervals.

**Figure 4 entropy-25-00301-f004:**
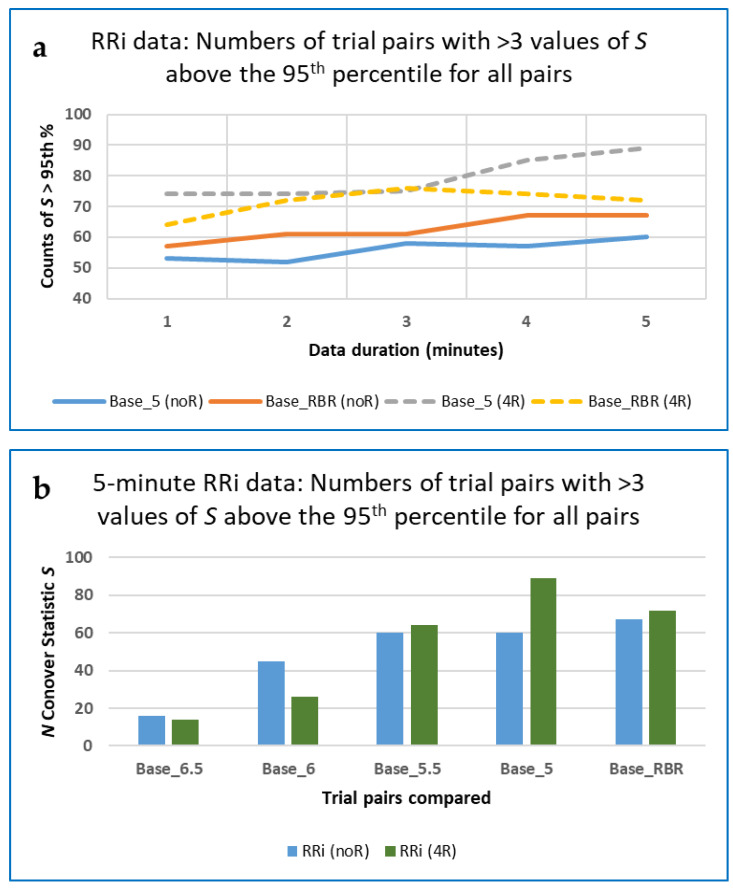
(**a**) RRi data: Numbers of Baseline to 5 BrPM (Ba_50) and Baseline to RBR (Ba_RBR) trial pairs with > 3 values of the Conover *S* statistic above the 95th percentile for all pairs, for all durations of data (1 to 5 min), for non-resampled (noR) and resampled (4R) data. The threshold of ‘> 3 values’ of *S* was selected because, for most comparisons, counts were very low (0 or 1), so that their upper quartile (75th percentile) was 4; (**b**) numbers of the five trial pairs with most values of the Conover *S* statistic above the 95th percentile for all pairs, for 5-min data only (no Baseline to 7 BrPM trial pairs met this criterion, nor did any Self-paced to externally paced trial pairs).

**Figure 5 entropy-25-00301-f005:**
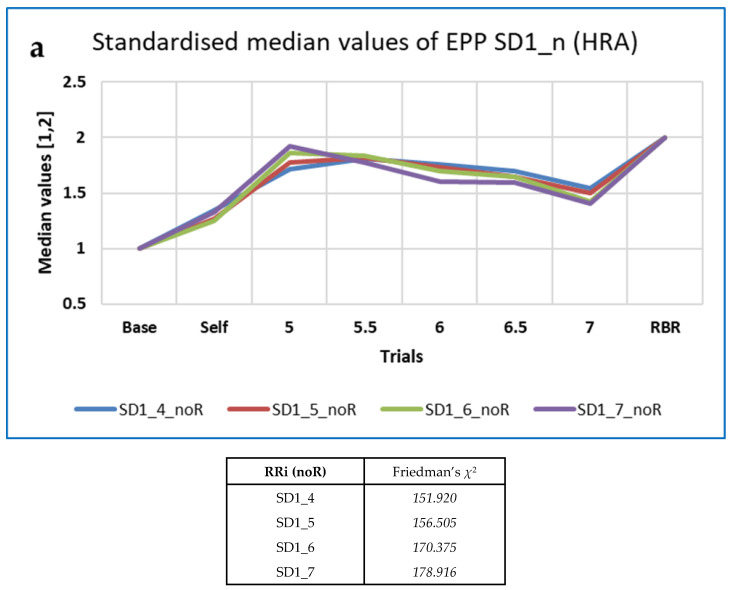
Examples of ‘Top’ CEPS RRi measures that *decrease* as breathing frequency increases, by family, with values of Friedman’s *χ*^2^: (**a**) EPP SD1_4 to SD1_7 (HRA); (**b**) LLE32-36 (OC); (**c**) HRV frequency domain measures, from Kubios HRV. Note that values have been standardised to the range (1, 2), for ease of comparison.

**Figure 6 entropy-25-00301-f006:**
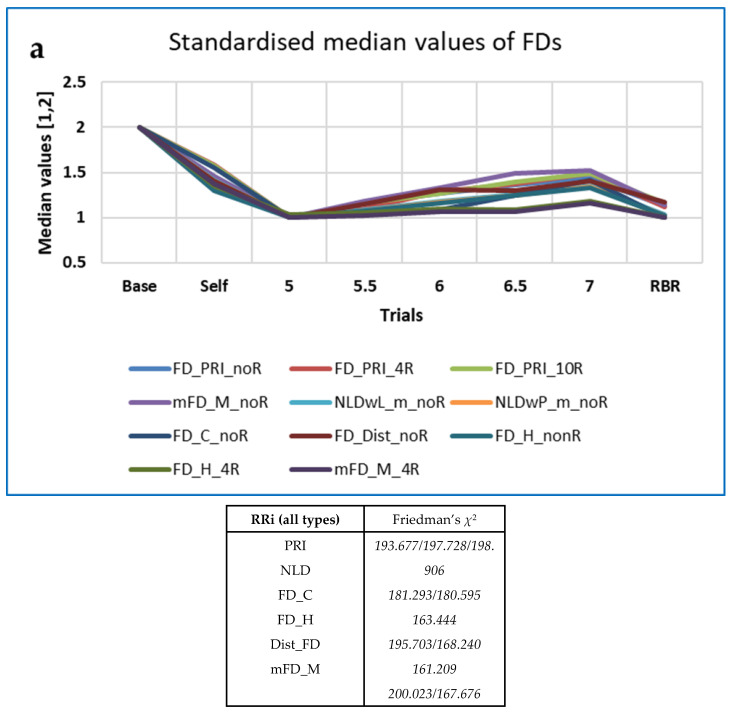
Examples of ‘Top’ CEPS RRi measures that *increase* as breathing frequency increases, by family, with values of Friedman’s χ^2^: (**a**) 11 FD measures (FD); (**b**) mPM_E and CPEI (PE-based); (**c**) Some other entropies (OE). Note that values have been standardised to the range (1,2), for ease of comparison, and that PJSC, a complexity rather than an entropy measure, decreases as breathing frequency increases.

**Figure 7 entropy-25-00301-f007:**
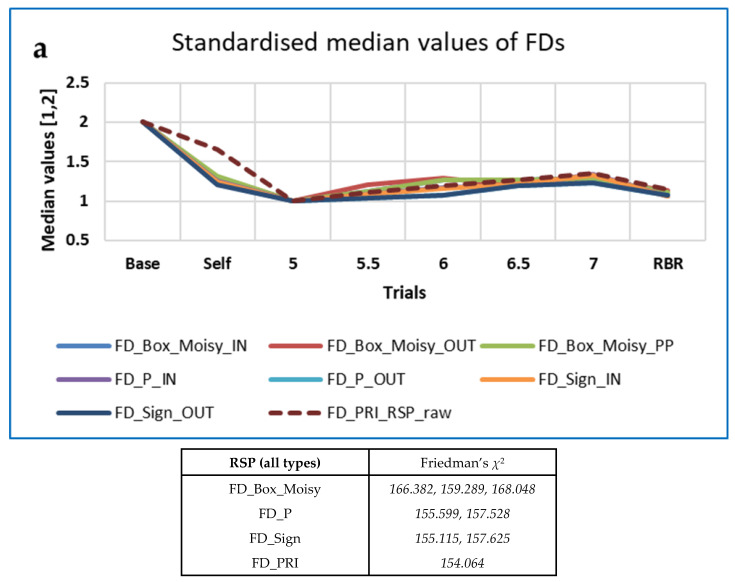
‘Top’ CEPS RSP measures, by family, with values of Friedman’s χ^2^. Note that values have been standardised to the range (1,2), for ease of comparison.

**Figure 8 entropy-25-00301-f008:**
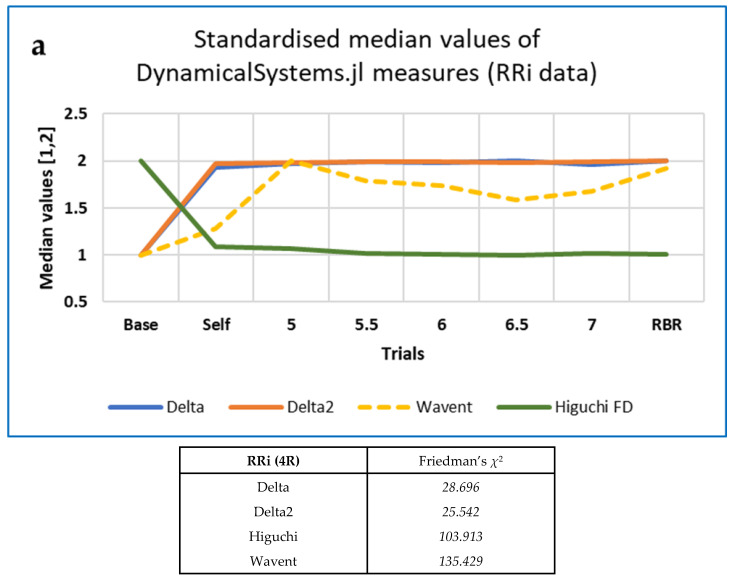
DynamicalSystems.jl FD measures, with Wavelet entropy (wavent) as a comparator: (**a**) 4R RRi data; (**b**) raw RSP data; (**c**) EDA data. Note that values have been standardised to the range (1,2), for ease of comparison.

**Figure 9 entropy-25-00301-f009:**
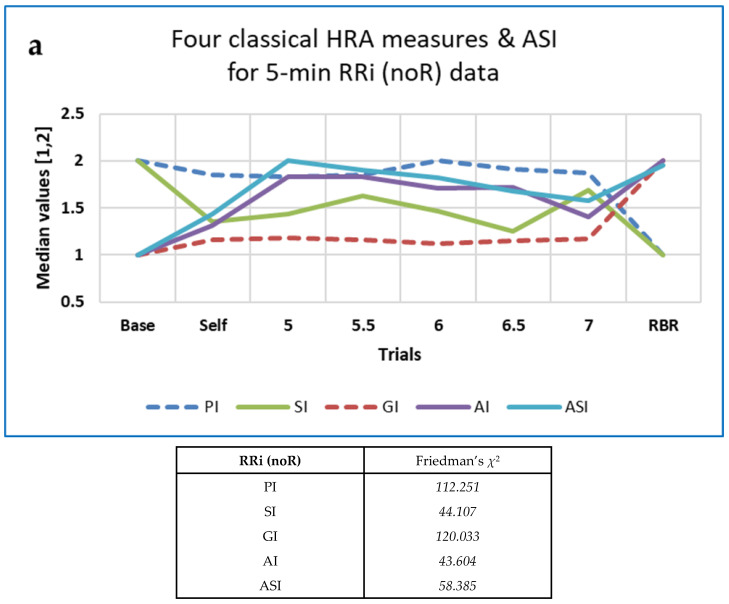
The effects of paced breathing on: (**a**) the ‘classical’ heart rate asymmetry (HRA) indices and the Asymmetric Spread Index (ASI); (**b**) Guzik’s subsidiary descriptors; (**c**) normalised HRA measures. Ehlers’ Index (Friedman’s *χ*^2^ = 92.926) is not shown.

**Figure 10 entropy-25-00301-f010:**
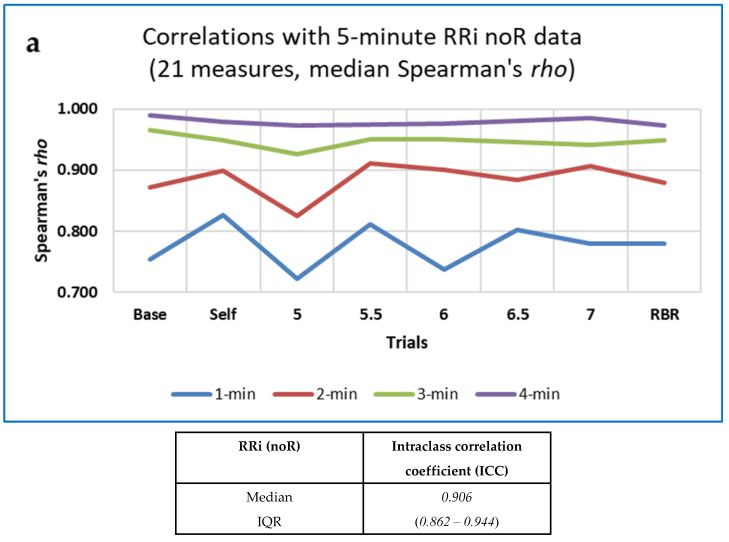
Left: Correlations (Spearman’s *rho*) between measures for 1-min, 2-min, 3-min or 4-min data with the same measures for 5-min data: (**a**) RRi noR data; (**b**) RRi (4R) data; (**c**) RRi (10R) data. Median and IQR of the ICCs for each of the three sets of measures are shown below each Figure part.

**Figure 11 entropy-25-00301-f011:**
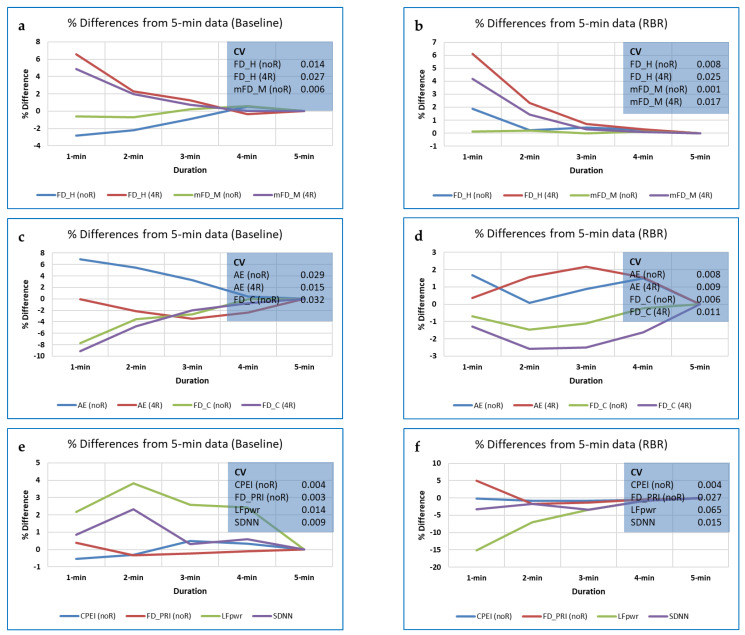
Percentage differences between median values of the measures for 1-min, 2-min, 3-min or 4-min data and the same measures for 5-min data. Top row (**a**,**b**): Two FD measures, for RRi (noR) and RRi (4R) data; Middle row (**c**,**d**): Average entropy (AE) and FD_C, for RRi (noR) and RRi (4R) data; Bottom row (**e**,**f**): Permutation-based CPEI and FD_PRI for non-resampled data, with RR-APET measures LF power and SDNN. Note that y-axes are all at different scales.

**Figure 12 entropy-25-00301-f012:**
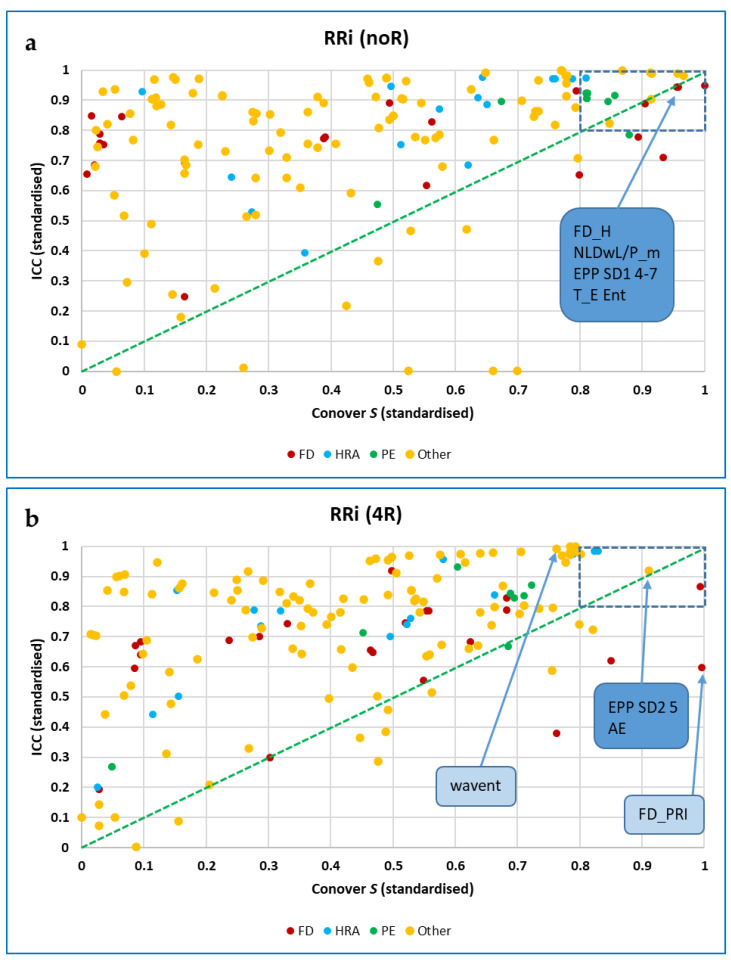
Scatter plots of standardised values of ICC and Conover *S*, for measures based on: (**a**) RRi (noR) measures; (**b**) RRi (4R) measures; (**c**) EDA measures, and (**d**) HRV measures from RR-APET. In parts (**a**–**c**), data points are in red for FDs, in blue for HRAs (in blue) and in green for PE-based measures. In part (**d**), points in red are for time-domain measures, points in blue for frequency-domain measures, in green for nonlinear complexity measures, and in yellow for the nonlinear RQA measures.

**Figure 13 entropy-25-00301-f013:**
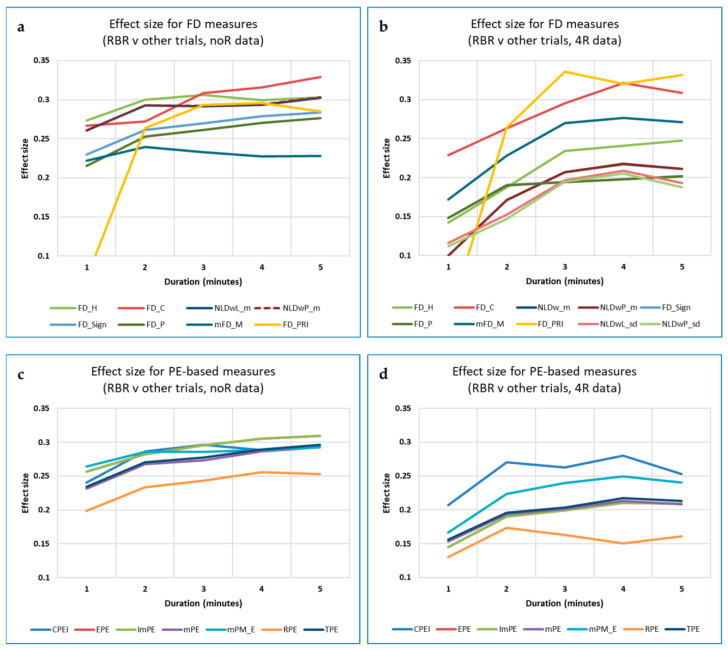
Changes in Mann-Whitney effect sizes (RBR vs. ‘rest’) with data length (1- to 5-min): (**a**,**b**) fractal dimensions; (**c**,**d**) PE-based measures; (**e**,**f**) some HRA indices; (**g**) RR-APET HRV measures (the Welch periodogram method was used for the frequency domain measures). Note that y-axes are not all to the same scale.

**Table 1 entropy-25-00301-t001:** A literature review of fractal dimension measures. Columns show Name, Abbreviation used for each measure, Selected references, numbers of studies located using PubMed and Google Scholar, and date of the first publication located for each measure that included the terms “fractal dimension” AND [Name]. These dates may not, however, indicate first publication of a particular FD measure. Numbers of hits for “[Name OR Name’s] fractal dimension” are shown in parentheses. All the measures listed, except those from Witold Kinsner, have been used in this paper. In this and the following Tables, alternating rows have been given a coloured background simply to aid readability.

Name	Abbrev.	Selected References	PubMed		Google Scholar	
			*N*	Date 1st	*N*	Date 1st
Higuchi	FD_H	Higuchi 1988 [14]	153 (116)	1994	5180 (1610)	1988
Katz	FD_K	Katz 1988 [15]	34 (16)	1994	6620 (436)	1985
Castiglioni	FD_C	Castiglioni 2010 [16]	3 (5)	2010	561 (13)	2010
Mandelbrot	FD_M	Castiglioni 2010 [16]	33 (42)	1975	46,400 (108)	1967
Petrosian	FD_P	Petrosian 1995 [17]	5 (8)	2010	876 (296)	1995
Sevcik	FD_S	Sevcik 1998 [18]	4 (4)	2009	534 (79)	1998
Box-count [Moisy]	FD_Box_M	Moisy 2022 [19]	370 (40)	1990	34,410 (2029)	c. 1985
Meerwijk/ van der Linden	FD_Box_MvdL	Meerwijk et al. 2015 [20]	2 (0)	2014	11 (0)	2015
Kalauzi	NLDwL NLDwP NLDiL NLDiP	Kalauzi et al. 2009 [8]	7 (0)	2005	243 (0)	2009
Tamulevičius, Kizlaitienė	FD_Amp FD_Dist FD_Sign FD_LRI FD_PRI	Kizlaitienė 2021 [21]	0 (0)	n/a	1 (0)	2021
Maragos	mFD_M	Maragos 1994; [22] Zlatintsi and Maragos 2013 [23]	2 (0)	1999	829 (4)	1993
Kinsner	D_β_ D_σ_	Kinsner 2008 [24]	1 (1)	2001	691 (2)	1989

**Table 2 entropy-25-00301-t002:** A literature review of heart rate asymmetry measures. Columns show Name, Abbreviation used for each measure, Selected references, numbers of studies located using PubMed and Google Scholar, and date of the first publication located for each measure that included the terms “heart rate asymmetry” AND [Name]. These dates may not indicate the first publication of a particular measure. Numbers of hits for “Name OR Name’s Index” are shown in parentheses. All the measures listed have been used in this paper.

Name	Abbrev.	Selected References	PubMed		Google Scholar	
			*N*	Date 1st	*N*	Date 1st
Ehlers’ Index	EI	Ehlers et al. 1998 [35]	4 (4)	2009	59 (37)	2006
Guzik’s Index	GI	Guzik et al. 2006 [36]	24 (9)	2006	1 (63)	2008
Porta’s Index	PI	Porta et al. 2006 [38]	15 (11)	2012	188 (123 ^a^)	2006
Slope Index (Karmakar)	SI	Karmakar et al. 2012 [39]	4 (4)	2015	28 (^b^)	2012
Area Index (Karmakar)	AI	Yan et al. 2017 [40]	3 (2 ^b^)	2017	17 (^b^)	2017
Asymmetric Spread Index (Rohila)	ASI	Rohila and Sharma 2020 [42]	0	n/a	2	2020
Deceleration contributions	SD1_up_, SD2_up_	Guzik et al. 2006 2006 [36]	1	2007	8, 0 (69, 12 ^b^)	2006
Acceleration contributions	SD1_down_, SD2_down_ ^c^	Guzik et al. 2006 2006 [36]	0	n/a	0, 0 (23, 12 ^b^)	n/a
SD1_up_^2^/SD1^2^, SD2_up_^2^/SD2^2^	C1_a_, C2_a_	Guzik et al. 2006 2006 [36] (adapted by Rohila)^d^	1	2022	5	2021
SD1_dn_^2^/SD1^2^, SD2_dn_^2^/SD2^2^	C1_d_, C2_d_	Guzik et al. 2006 2006 [36] (adapted by Rohila)	1	2022	5	2021
√((SD1_up_^2^ + SD2_up_^2^)/2)	SDNN_up_	Piskorski and Guzik 2012 [48]	1	2022	17	2021
√((SD1_down_^2^ + SD2_down_^2^)/2)	SDNN_down_	Piskorski and Guzik 2012 [48]	1	2022	17	2021

Notes. ^a^. There were 164 hits for ‘Porta Index”, and 41 for “La Porta Index”; ^b^. Terms such as “Slope Index” and “Area Index” are used in many contexts, so that searching for them without some form of qualifier such as author Name was not helpful; ^c^. Some authors now use SD1_low_ and SD2_low_ instead of SD1_down_ and SD2_down_ [49]; d. Guzik et al. originally used C_up_ and C_dn_ for SD1_up_^2^/SD1^2^ and SD1_dn_^2^/SD1^2^. The corresponding indices for SD2 were added in a later paper, along with the change to C_a_ and C_d_ [48]. We have used SDNN_up_ and SDNN_down_ rather than SDNN_a_ and SDNN_d_. In this and the other Tables in this paper, alternating rows have been given a coloured background simply to aid readability.

**Table 3 entropy-25-00301-t003:** Number of participants in each Age and Sex band.

Age	Female	Male	All
18–24	2	4	6
25–34	3	6	9
35–44	6	4	10
45–54	3	5	8
55–64	3	3	6
65–74	0	3	3
75–84	1	1	2
Total	18	26	44

**Table 4 entropy-25-00301-t004:** Other measures newly implemented in CEPS 2, or in course of implementation (asterisked). Measures planned for future inclusion are listed in parentheses (for measures already included in CEPS, see [4]). Measures are listed in alphabetical order, showing original references, names of code providers, code type and institutions of originators. Please note that, although every effort has been made to implement these measures correctly in CEPS, time has not always allowed us to validate the results obtained when using CEPS with those researchers who provided us with code. As with all Creative Commons licensing, CEPS is provided freely and without warranty, on condition that this paper is referenced in any outputs that result from using the software.

Measure	Original Author/s	Provider	Source Code	Institution
AttnEn	Yang et al. 2020 [80]	EntropyHub	MATLAB	Xi’an
AvgApEnP	Udhayakumar et al. 2017 [81]	Karmakar	MATLAB	Melbourne
AvgSampEnP	Udhayakumar et al. 2018 [82]	Karmakar	MATLAB	Melbourne
(B_ApEn)	Manis and Sassi 2021 [83]	Published paper	Python	Ioannina/ Milano
(B_SampEn)	Manis and Sassi 2021 [83]	Published paper	Python	Ioannina/ Milano
(C0)	Shen et al. 2005 [84]	(Panday)	tbc	Fudan
CAFE	Girault and Humeau-Heurtier 2018 [85]	Girault	MATLAB	Angers
CI *	Costa et al. 2008 [86]	Panday	MATLAB	Harvard
(CID)	Batista et al. 2013 [87]	Published paper	MATLAB	California (Riverside)
CmSE	Wu et al. 2013 [88]	Published paper	MATLAB	Taipei
CoSEn	Lake 2011 [89]	Liu	MATLAB	Virginia (Charlottesville)
CoSiEn	Chanwimalueang and Mandic 2017 [90]	EntropyHub	MATLAB	Imperial (London)
CPEI	Olofsen et al. 2008 [91]	Published paper	MATLAB	Leiden/ Auckland
DE *	Grigolini et al. 2001 [10]	Culbreth	MATLAB	North Texas (Denton)
DFA Alpha	Kugiumtzis and Tsimpiris 2010 [92]	Published paper	MATLAB	Thessaloniki
DiffEn *	Shi et al. 2013 [93]	(Panday)	MATLAB	Shanghai
(EE)	Giannakopoulos and Pikrakis [94]	Mathworks	MATLAB	Agia Paraskevi
EPE	Huo et al. 2019 [95]	Huo	MATLAB	Lincoln
ESCHA *	Fernández et al. 2014 [96]	Santamaría Bonfil	R	CONACYT-INEEL, Cuernavaca
FastLomb *	Scargle 1982 [97]	Mathworks	MATLAB	California (Berkeley)
FFT *	Cooley and Tukey 1965 [98]	Mathworks	MATLAB	IBM, New York
GPP *	Platiša et al. 2022 [9]	Kalauzi	MATLAB	Belgrade
GridEn	Yan et al. 2019 [99]	EntropyHub	MATLAB	Shandong
IncrEn	Liu et al. 2016 [100]	EntropyHub	MATLAB	Changzhou
Jitter_Jitt	Teixeira et al. 2013 [101]	Teixeira	MATLAB	Bragança
Jitter_Jitta	Teixeira et al. 2013 [101]	Teixeira	MATLAB	Bragança
Jitter_ppq5	Teixeira et al. 2013 [101]	Teixeira	MATLAB	Bragança
Jitter_RAP	Teixeira et al. 2013 [101]	Teixeira	MATLAB	Bragança
L_ApEn *	Manis and Sassi 2021 [83]	Published paper	Python	Ioannina/ Milano
L_SampEn *	Manis and Sassi 2021 [83]	Published paper	Python	Ioannina/ Milano
LZPC *	Zozor et al. 2014 [102]	GitHub	C	Grenoble/ Córdoba
MESA *	Burg 1975 [103]	Dowse	MATLAB	Stanford
mFmDFA *	Castiglioni and Faini 2019 [104]	Castiglioni	MATLAB	Milano
MmSE	Wu et al. 2013 [105]	Published paper	MATLAB	Taipei
mPhEn	Panday n.p.	Panday	MATLAB	Hertfordshire
PJSC	Zunino et al. 2012 [106]	Zunino	MATLAB	La Plata
PLZC *	Bai et al. 2015 [107]	Published paper	MATLAB	Yanshan
QSE *	Lake 2011 [108]	(Panday)	MATLAB	Virginia (Charlottesville)
(RPDE)	Little et al. 2007 [109]	GitHub: hctsa	MATLAB	Oxford
RPE	Jauregui et al. 2018 [110]	Zunino	MATLAB	Maringá
SEx	Lad et al. 2015 [111]	Sanfilippo/Panday	MATLAB	Canterbury, NZ/Palermo
Shimmer_Shim	Teixeira et al. 2013 [101]	Teixeira	MATLAB	Bragança
Shimmer_ShdB	Teixeira et al. 2013 [101]	Teixeira	MATLAB	Bragança
Shimmer_apq3	Teixeira et al. 2013 [101]	Teixeira	MATLAB	Bragança
Shimmer_apq5	Teixeira et al. 2013 [101]	Teixeira	MATLAB	Bragança
SpEn	Inouye et al. 1991 [112]	Mathworks	MATLAB	Osaka
SQA *	Girault 2015 [113]	Girault	MATLAB	Angers
SymDyn *	Various (see Primer)	(Panday)	MATLAB	Various
(Tangle)	Moulder et al. 2022 [114]	GitHub	R	Virginia (Charlottesville)
TPE	Zunino et al. 2008 [115]	Zunino	MATLAB	La Plata
VM *	Bernaola- Galván et al. 2017 [116]	Bernaola- Galván/ Panday	Fortran	Málaga
(WE)	Rosso et al. 2001 [117]	Mathworks	MATLAB	Buenos Aires

**Table 5 entropy-25-00301-t005:** The 95th percentile of the Conover statistic for six groups of measures (5-min RRi data).

Data Type	95%/*N*	FD	HRA	PE	RQA	OC	OE	ALL
noR	95th %	11.348	9.927	11.908	5.223	6.761	7.178	8.838
	*N*	22	40	8	17	51	54	192
RRi 4R	95th %	9.697	8.848	9.971	5.864	5.267	8.831	9.038
	*N*	22	40	10	19	51	54	196
RRi 10R	95th %	8.288	9.566	9.384	6.229	9.414	9.262	9.089
	*N*	22	40	8	19	48	51	188

**Table 6 entropy-25-00301-t006:** 5-min RRi data: median values of Friedman’s *χ*^2^ and Kendall’s *W* for six groups of CEPS measures, with interquartile ranges (IQRs) in parentheses.

Data Type	*χ*^2^/*W*	FD	HRA	PE	RQA	OC	OE	ALL
noR	*χ* ^2^	96.597	106.888	139.596	28.953	55.626	50.863	62.795
	*W*	0.310	0.350	0.448	0.093	0.178	0.163	0.202
RRi 4R	*χ* ^2^	78.813	92.505	114.482	33.286	103.706	86.055	78.813
	*W*	0.253	0.297	0.367	0.103	0.333	0.276	0.253
RRi 10R	*χ* ^2^	63.412	113.907	108.155	38.291	102.892	103.603	95.832
	*W*	0.203	0.365	0.347	0.123	0.330	0.332	0.307

**Table 7 entropy-25-00301-t007:** Results for the best-performing RRi measures (values of χ^2^ > 150), with measures from Kubios HRV provided as a comparison in the lower part of the Table.

Data Type	Measures and *χ*^2^ Range	FD	HRA	PE	RQA	OC	OE	Best
noR	Measures	mFD_M FD_PRI NLDw_mL NLDw_mP FD_C FD_Dist	EPP SD1 (4–7)	CPEI mPM_E PJSC	n/a	n/a	T_E_Ent	mFD_M
	*χ*^2^ range	161.208–200.023	151.920–178.916	150.519–161.949			173.660	200.023
RRi 4R	Measures	FD_PRI FD_H mFD_M	n/a	n/a	n/a	LLE32–36	FE MmSE2, MmSE5 AE	FD_PRI
	*χ* ^2^	167.676–197.728				156.084–170.126	150.084–158.552	197.728
RRi 10R	Measures	FD_PRI	n/a	n/a	n/a	n/a	BE	FD_PRI
	*χ* ^2^	198.906					158.334	198.906

**Kubios HRV**		**General**	**HRA**	**Time domain**	**Freq domain**	**OC**	**OE**	**Best**
RRi 4R	Measures	PLFP	SD2 SD2/SD1	SDNN	LFpwr (AR/LS) Totpwr (AR/LS)	DFA alpha1	SampEnApEn	PLFP [AR LFpwr]
	*χ* ^2^	201.714	164.605–165.441	155.624	154.964–181.803	167.895	155.865–157.280	201.714 [181.803]

**Table 8 entropy-25-00301-t008:** CEPS measures for the 5-min RRi data with standardised values of Conover S for the Baseline-RBR pair ≥ 0.8 are shown below, with non-standardised values of Conover S in parentheses. In **bold**, values of Conover S for those measures with standardised ICC also ≥0.8. RRi 10R measures with numbers of increases or decreases < 35 are not included. ‘↑’ indicates measure increased between baseline and RBR, and ‘↓’ that it decreased, for the number of participants included in parentheses.

5-min RRi	Measures	noR	4R	10R
Baseline-RBR ↑	SDNNdown SD2down PJSC T_E_ENT EPP SD1_3 EPP SD1_4 EPP SD1_5 EPP SD1_6 EPP SD1_7 EPP SD2_5 AE	**0.809** (12.829) (43 ↑) 0.879 (13.934) (44 ↑) **0.914** (14.483) (43 ↑) **0.868** (13.758) (43 ↑) **0.913** (14.470) (43 ↑) **0.915** (14.502) (43 ↑) **0.956** (15.151) (43 ↑) **0.966** (15.310) (43 ↑)	**0.824** (12.733) (43 ↑) **0.830** (12.828) (43 ↑) **0.801** (12.392) (43 ↑) **0.955** (14.084) (42 ↑)	0.806 (12.624) (43 ↑) 0.806 (12.624) (43 ↑) (6.796) (35 ↑) (7.899) (39 ↑) (11.964) (43 ↑) (9.074) (39 ↑)
Baseline-RBR ↓	FD_C FD_H mFD_M FD_PRI EPP R6 CPEI EPE ImPE mPE1 mPM_E TPE FE MmSE2	0.893 (15.151) (44 ↓) **1.000** (15.841) (44 ↓) **0.904** (14.327) (43 ↓) 0.933 (14.790) (44 ↓) **0.847** (13.423) (43 ↓) **0.845** (13.387) (44 ↓) **0.811** (12.859) (43 ↓) **0.811** (12.859) (43 ↓) **0.812** (12.877) (43 ↓) **0.856** (13.562) (43 ↓) **0.809** (12.826) (43 ↓)	0.851 (13.151) (44 ↓) **0.994** (15.356) (44 ↓) **1.000** (15.453) (44 ↓) 1.000 (15.392) (44 ↓) 0.800 (12.364) (44 ↓) 0.821 (12.693) (43 ↓)	(7.534) (37 ↓) (10.150) (43 ↓) (12.106) (44 ↓) 1.000 (15.541) (44 ↓) (8.641) (40 ↓) (10.304) (41 ↓) (9.903) (42 ↓) (9.903) (42 ↓) (10.301) (43 ↓) (8.820) (40 ↓) (10.59) (42 ↓) (4.380) (35 ↓) (9.609) (41 ↓)

**Table 9 entropy-25-00301-t009:** Numbers of RSP data measures for which Friedman’s χ^2^ > 150.

Data Type	Measures and *χ*^2^ Range	FD	HRA	PE	RQA	OC	OE	Best
INbreath	Measures	3	n/a	7	n/a	18	18	MmSE13
	*χ*^2^ range	155.115–166.382		154.082–187.089		150.490–190.092	159.045–192.043	192.043
OUTbreath	Measures	3	7	7	1	17	19	MmSE13
	*χ* ^2^	157.528–159.289	151.338–167.358	160.749–188.632	167.753	151.639–188.884	150.950–195.610	195.610
Peak-Peak (PP)	Measures	1	8	7	n/a	17	22	ImPE
	*χ* ^2^	168.048	177.494–184.878	171.671–191.663		150.569–191.249	152.381–190.565	191.663

Raw RSP	Measures	FD_PRI	n/a	n/a	n/a	n/a	n/a	FD_PRI
	*χ* ^2^	154.064						154.064

**Table 10 entropy-25-00301-t010:** Top two Friedman test results for differences in CEPS measures among all eight trials, for the RRi, RSP and EDA data, with corresponding results for the Kubios HRV measures.

Data Type	Measures	Friedman’s *χ*^2^	Kendall’s *W*
RRi (noR)	mFD_M FD_H	200.023 195.703	0.642 0.628
RRi (4R)	FD_PRI LLE34	197.728 170.126	0.633 0.546
RRi (10R)	FD_PRI BE	198.906 158.334	0.637 0.508
RSP (IN)	LLE42 EoD	190.092 188.255	0.251 0.600
RSP (OUT)	LLE43 TPE	188.884 188.642	0.250 0.605
RSP (PP)	ImPE ESCHA_d	191.663 191.249	0.613 0.612
RSP (Raw)	FD_PRI FD_LRI	154.064 122.040	0.493 0.389
EDA	RMSSD EPP SD1_1	29.035 28.824	0.093 0.924
Kubios HRV	PLFP LFpwr (AR)	201.714 181.803	0.647 0.583

Note: MmSE13 for RSP (IN, OUT) was not included in this summary (see above).

**Table 11 entropy-25-00301-t011:** Top two Friedman test results for differences in DynamicalSystems.jl measures among all eight trials, for the RRi (4R), deduplicated Raw RSP and deduplicated detrended EDA data.

Data Type	Measures	Friedman’s *χ*^2^	Kendall’s *W*
RRi (4R)	Wavent Perment4	135.429 118.979	0.440 0.386
RSP (Raw)	Wavent Perment4	66.910 51.835	0.217 0.168
EDA	Wavent Delta2	16.609 15.214	0.054 0.049

**Table 12 entropy-25-00301-t012:** ‘Top five’ non-standardised values of Conover *S* for the ECG RRi data, for all 28 pairs of trials.

4R 5-min	Pair	*S*	10R 5-min	Pair	*S*	NoR 5-min	All Base_5
FD_PRI	Base_5	19.013	FD_PRI	Base_5	19.508	mFD_M	19.334
FD_PRI	Base_5.5	15.826	FD_PRI	Base_5.5	15.614	FD_PRI	19.163
mFD_M	Base_RBR	15.453	FD_PRI	Base_RBR	15.541	FD_H	18.672
FD_PRI	Base_RBR	15.392	MmSE11	Base_5	13.933	NLDwL_m	16.566
FD_H	Base_RBR	15.356	MmSE10	Base_5	13.888	NLDwP_m	16.521
**Medians**		15.453			15.541		18.672

**Table 13 entropy-25-00301-t013:** ‘Top five’ non-standardised values of Conover S for the breathing interval data, for all 28 pairs of trials.

IN 5-min	All Base_5	OUT 5-min	All Base_5	PP 5-min	All Base_5
IncrEn	18.037	ImPE	17.907	ImPE	18.302
EoD	17.898	Discrete_CS	17.873	Discrete_CS	18.224
KLD	17.898	IncrEn	17.867	EoD	18.096
ImPE	17.849	TPE	17.833	KLD	18.096
Discrete_CS	17.713	mPM_E	17.794	mPM_E	18.057
**Medians**	17.898		17.867		18.096

**Table 14 entropy-25-00301-t014:** ‘Top five’ non-standardised values of Conover S for the raw respiration (RSP) and EDA data, for all 28 pairs of trials.

RSP 5-min	Pair	*S*	EDA 5-min	Pair	*S*
FD_PRI	Base_5	14.349	GridEn	Base_6	5.250
FD_PRI	Base_RBR	12.08	Jitta	Base_5	4.829
FD_PRI	Base_5.5	11.620	RMSSD	Base_5	4.730
FD_PRI	7_5	9.627	EPP SD1_1	Base_5	4.728
FD_LRI	Base_5	11.091	EPP SD1_2	Base_5	4.728
**Medians**		11.620			4.730

**Table 15 entropy-25-00301-t015:** Median values of Conover *S* for four pairs of trials, with lowest values in each row indicated by bold type.

Conover *S*	Self to RBR	Base to RBR	Self to 5 BrPM	Base to 5 BrPM
RRi (4R) (225)	4.601	7.386	**4.091**	7.129
RRi (10R) (209)	5.267	8.536	**5.007**	8.538
RRi (noR) (219)	3.673	6.105	**3.182**	6.084
RSP raw (99)	**2.549**	3.307	3.069	3.277
RSP_IN (196)	**2.770**	7.507	4.270	8.723
RSP_OUT (197)	**4.250**	8.070	4.676	9.094
RSP_PP (197)	**4.180**	7.794	5.389	9.548
EDA (89)	**0.812**	1.732	0.892	1.865

**Table 16 entropy-25-00301-t016:** Measures with standardised median values of *χ*^2^, Kendall’s *W* and the Baseline-RBR Conover statistic *S* > 0.8, for two RRi data types and RR-APET. Measures with values lower than any one of these thresholds for one or more of the 2-, 3-, 4- or 5-min data segments are not included. Values within each cell are in order (top to bottom) *χ*^2^, *W* and *S*. Total numbers of measures analysed for each data type are shown in parentheses.

RRi (noR) (219)		RRi (4R) (224)		RR-APET (25)	
mFD_M	194.4 0.624 13.327	SD2down	185.6 0.595 12.828	SD2	139.7 0.448 12.319
FD_H	187.8 0.602 15.239	mFD_M	156.8 0.503 13.562	SDNN	136.7 0.439 12.132
FD_PRI	187.7 0.601 14.790			Alpha1	132.8 0.426 10.251
EPP SD1_7	176.2 0.565 14.691				
NLDwL_m	172.1 0.552 14.435				
NLDwP_m	171.8 0.551 14.541				
EPP SD1_6	170.7 0.548 15.300				
FD_C	161.8 0.519 13.508				
CPEI	157.0 0.504 13.043				

**Table 17 entropy-25-00301-t017:** ‘Top 13′ measures for each RRi data type, with most differences of the same sign across the 1-, 2-, 3-, 4- and 5-min data, together with the ‘top 4′ HRV indices. Counts are of increases (↑) or decreases (↓) between Baseline and RBR. Counts excluding the 1-min data are shown in parentheses. Measures for which ICC > 0.9 are in bold type, and total numbers of measures analysed for each data type are shown in parentheses.

RRi (noR) (220)		RRi (4R) (220)		RR-APET (24)	
**PJSC** (↑)	**199 (162)**	**FD_C** (↓)	191 (157)	**SD2** (↑)	215 (172)
**RoCV** (↑)	191 (154)	mFD_M (↓)	191 (155)	SDNN (↑)	214 (172)
**EPP SD1_6** (↑)	188 (150)	**AE** (↑)	191 (154)	Alpha1 (↑)	213 (171)
ACR5 (↓)	187 (151)	RCmDE7 (↓)	190 (156)	LFpwr (↑)	211 (170)
**EPP SD1_5** (↑)	187 (149) ^a^	**Q3** (↑)	188 (153)		
**mPE** (↓)	186 (153)	FD_H (↓)	187 (152)		
EPP r5 (↓)	186 (150)	LLE32 (↑)	187 (150)		
**EPE** (↓)	185 (152)	LLE33 (↑)	187 (149)		
**ImPE** (↓)	185 (152)	**RoCV** (↑)	185 (149)		
**AE** (↑)	185 (149) ^a^	LLE31 (↑)	183 (146)		
EoD (↓)	184 (151)	RCmDE6 (↓)	182 (151)		
KLD (↓)	184 (151)	**SD2_down_** and **SDNN_down_** (↑)	182 (146)		
**MPM_E** (↓)	184 (149)	LLE30 (↑)	182 (145)		

^a^. EPP SD1_5 and AE were not in the ‘top 13′ of counts excluding the 1-min data.

**Table 18 entropy-25-00301-t018:** Measures with short-data values consistently within ± 5% of their values for the 5-min data, with those measures least often within ± 5% of their values for the 5-min data.

Top 12 Measures	ICC	Median CV	Count
FD_H (NoR)	0.947	0.003	32
NLDwL_m (NoR)	0.944	0.001	32
NLDwP_m (NoR)	0.944	0.001	32
Q3 (4R)	0.910	0.008	32
CPEI (NoR)	0.897	0.008	32
mFD_M (4R)	0.894	0.016	32
mFD_M (NoR)	0.889	0.003	32
LLE33 (4R)	0.767	0.011	32
LLE32 (4R)	0.747	0.011	32
Alpha1	0.823	0.018	31
LLE30 (4R)	0.737	0.012	31
LLE31 (4R)	0.730	0.014	31
Bottom five measures	ICC	Median CV	Count
PJSC (NoR)	0.786	0.038	15
EoD (NoR)	0.849	0.069	11
KLD (NoR)	0.849	0.069	11
ACR5 (NoR)	0.864	0.124	2
EPP R5 (NoR)	0.862	0.138	2

## Data Availability

The data presented in this study and from our previous study will shortly be freely available in Open Research Data Online (ORDO), The Open University’s searchable research data repository at https://ordo.open.ac.uk/ (accessed on 27 November 2022).

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
