# Peer review of "Complexity and Entropy in Physiological Signals (CEPS): Resonance Breathing Rate Assessed Using Measures of Fractal Dimension, Heart Rate Asymmetry and Permutation Entropy"

_entropy, 2023, doi:10.3390/e25020301_

Round 1
Reviewer 1 Report
The authors present impressive work regarding the development of their CEPS toolbox. Their goal is quite ambitious, regarding the number of implemented parameters that were chosen based on comprehensive literature research. The toolbox and the paper will be clearly useful for the field, and a major strength of the study is that it can be adequately reproduced as methods were described in detail.
I am concerned about the followings:
- figures do not display statistics (like SD, SEM and significance), which must be corrected
- a clear hypothesis should be presented, were there any, and if yes, what were the a priori expectations regarding behaviors of the parameter,
- the paper provides a lot of unnecessary details about how the work was developed (communications with co-authors and potential collaborators - it is not of general interest who did not share their code), and also long word-by-word citation. For example, fractal dimension should be defined in an explicit precise manner by using mathematical formulae.
- multifractal parameters: a clear definition is also necessary as analysis of multifractality yields a set of scaling exponents, and endpoints typically reflect the width of distribution and characteristic values of such set
The following aspects should be considered for toolbox development and in a paper providing demonstration of a toolbox it is useful to recommend best practices regarding:
- surrogate testing procedures for model-specific measures as fractal analysis assumes a scale-free model for the data that needs to be tested, especially on physiological processes with strong oscillatory component
- applications would benefit from real-time implementation as it allows for monitoring and the underlying algorithms are optimized for low computational cost. Efficient calculation is very important practical aspect for future users of the toolbox. See an example paper here and other papers from the group
Hartmann, A. et al. Real-time fractal signal processing in the time domain. Physica A 392, 89-102, doi:10.1016/j.physa.2012.08.002 (2013).
- Another interesting application would be to measure the synchronization between breathing and heart rate:
Bartsch RP, Liu KKL, Ma QDY, and Ivanov PCh. Three independent forms of cardio-respiratory coupling: transitions across sleep stages. Computing in Cardiology, 2014; 41:781-784
Reviewer 2 Report
In this study the Authors present a wide exploration of an improved/extended version of their previously developed software package, CEPS. Overall, the work done is substantial, precise and of high quality. Ironically, however, the manuscript has one severe flaw, in my opinion: its complexity is beyond that of with what a general audience could (or likely would) cope with. The manuscript is overly long, utilizes an immense set of mathematical measures for data analysis, and explores many aspects (such as effect of data length, redundancy, discriminability and so on) for most of them, which makes the whole manuscript quite hard to follow and comprehend. I feel that this is in contrast with he original goal of the development of the CEPS package in the first place, which was designed to provide an easy-to-use platform for clinicians/medical researchers who are not necessarily highly skilled in computer programming and/or mathematics. Nevertheless, I think the work done throughout this project is admirable and worth of publishing. I will provide some points/suggestions below that might help the Authors further improve their manuscript before acceptance. Note that these will be listed rather in order of appearance and not that of importance.
- I think the abstract does not necessarily reflect the main concept of the paper itself. As highlighted at the end, ' Conclusions: The updated CEPS software enables multichannel physiological data to be visualised using a variety of complexity and recently introduced entropy measures. The software could be adapted for continuous patient monitoring in critical care and may have the potential to transform healthcare management in the future, particularly during epidemics.'. Therefore, I feel it unnecessary to provide a detailed report of how many individual indices computed changed in response to paced breathing, instead a wholistic approach in the abstract would be preferred, summarizing the main tendency of the results broken down as parameter families (e.g., measures of fractal dimension, entropy, etc. expressed a tendency of changing/being unaffected by respiratory rate). This would help to shorten the abstract and highlight the key goal of the study instead of focusing on specific differences observed.
- I understand that with 200+ abbreviations spread over 70+ pages it is very difficult to maintain consistency, however please double-check that all abbreviations are defined first, and then only the abbreviation is used in following appearances. Some abbreviations are introduced more, than once (e.g., CEPS and ASI is introduced twice, HRA is defined three times only on page 5), while others are not introduced at all (such as SD1/SD2, DFA).
- I understand that providing an exact definition for this many parameters is impossible within the scope of the paper, however I think it would be useful to provide definitions for concepts such as time irreversibility or heart rate asymmetry.
- In general I think the biggest shortcoming of the paper is its length. Given that most of the text is devoted to exploring the parameters and their affectedness by the paced breathing paradigm, I would reconsider if some parts of the manuscript could be moved to supplementary material from the main text. The text often feels more like an internal study report rather than an original research article, often explicitly reflecting the Authors' personal viewpoints or the challenges/difficulties they were facing while conducting the study. I would consider collecting these topics (such as detailing Covid risk mitigation, personal challenges, segment 4.3 anxieties of data collection and so on) - that did not contribute to the main scientific content of the study - into a separate segment in the supplementary material, so not to burden the main text that is already very long.
- I understand that the main goal of this study was to demonstrate the improved CEPS package, explore the various implications one can derive from its plethora of available parameters, and compare its performance with other packages out there. However, from the discussion I miss the actual physiological interpretations/implications one can derive from the applied study protocol itself. I think this aspect is of high importance, given that the Authors state as a goal/study conclusion that 'The software could be adapted for continuous patient monitoring in critical care and may have the potential to transform healthcare management in the future, particularly during epidemics.'. For example, what physiological regulatory process does it indicate that fractal dimension- or permutation entropy-related measures change with various breathing rates? In what clinical conditions these measures can be useful and why? For critical care patient monitoring, which of these measures can be computed in real time or on a relevant time scale of what medical condition? In my opinion strengthening these aspects would help maintain the investment of the reader, especially if the main target audience is medical practitioners (among others).
Some other minor points:
- In many of the figures (e.g., Fig. 7, Fig. 8, Fig. 20) the exact size of the panels and/or their spacing is uneven.
- In the Discussion, the sentences starting with 'Reluctantly, we were forced to conclude, with [31], that Procrustean binary thinking...' are included twice, once in segment 4.1 and then again in segment 4.5.
- For future directions/further expansion of the CEPS package, the authors might want to include some other parameters that can be of use in characterizing the temporal complexity of univariate/bivariate physiological signals, such as multifractal measures derived via focus-based multifractal formalism (Mukli et al., 2015), or real-time detrended cross-correlation (Kaposzta et al., 2022). Multifractal measures of heart rate variability can be of particular interest regarding that they can be affected in physiological and pathological conditions (see e.g., Ivanov et al. 1998, Amaral et al. 2001).
- The Authors might also be interested in the recently (re-)introduced concept of network physiology (e.g., Bashan et al., 2012, Bartsch et al., 2015) focusing not only on the complexity in the temporal dynamics of individual organ systems (such as heart rate, respiratory rate, etc.), but also how they dynamically interact. Therefore, it could be considered to include bi-/multivariate measures of complexity into the CEPS package to analyze multichannel physiological recordings. This might provide further insight on how to interpret changes in complexity measures from multichannel recordings in response to specific paradigms (such as paced breathing at different rates). For example, it has been shown that the phase coupling of heart and respiratory rates can exist in multiple forms depending on sleep stage (Bartsch et al., 2015).
Round 2
Reviewer 2 Report
Thank you for addressing my points!